# Conformal Prediction for Early Stopping in Mixed Integer Optimization

Stefan Clarke [1]   Bartolomeo Stellato [1]

## Abstract

Mixed-integer optimization solvers often find optimal solutions early in the search, yet spend the majority of computation time proving optimality. We exploit this by learning when to terminate solvers early on distributions of similar problem instances. Our method trains a neural network to estimate the true optimality gap from the solver state, then uses conformal prediction to calibrate a stopping threshold with rigorous probabilistic guarantees on solution quality. On six problem families from the Distributional MIPLIB library, our method reduces solve time by over 60% while guaranteeing 0.1%-optimal solutions with 95% probability for new instances drawn from the same distribution.

## 1. Introduction

Mixed-integer optimization problems arise in robotics (Shin et al., 2022; Halsted et al., 2021), transportation (Toth & Vigo, 2002; Bertsimas et al., 2019), power systems (Hentenryck, 2021; Sarkar et al., 2018), and control (Marcucci & Tedrake, 2019). Many applications require decisions in real time, yet mixed-integer programming is NP-hard (Garey & Johnson, 1979), and even modern solvers (Gurobi Optimization, LLC, 2024; Huangfu & Hall, 2018) may be too slow for time-critical settings.

The standard method for solving mixed-integer programs is branch-and-bound (Land & Doig, 1960), often combined with cutting planes to form branch-and-cut (Mitchell, 2002). As the algorithm progresses it maintains upper and lower bounds on the optimal objective value; when these bounds meet, the solver terminates with a provably optimal solution. Heuristics help find good feasible solutions early (Fischetti & Lodi, 2011; Angioni et al., 2025), quickly reducing the upper bound, but closing the lower bound to certify optimality typically requires many more iterations. The result

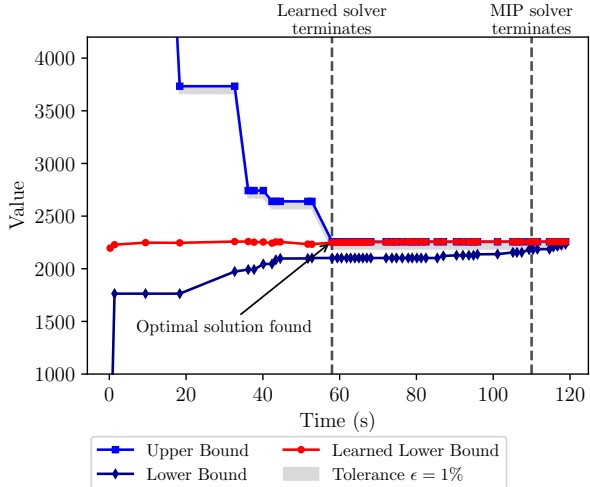

*Figure 1.* Bounds for one solve of one instance of the Distributional MIPLIB optimal transmission switching problem family (Huang et al., 2024). The optimal solution is found after about 57 seconds, but the solve does not terminate until nearly 120 seconds have passed because the gap is not within the tolerance. Our method with a learned lower-bound terminates the solve much earlier, around 57 seconds. The upper-bound remains large until around 20 seconds, after which it quickly decreases as the solver finds feasible solutions.

is that the solver often discovers an optimal solution long before it can prove optimality (see Figure 1).

Machine learning has emerged as a promising tool to accelerate mixed-integer optimization. Bengio et al. (2021) survey the landscape, while foundational work includes learning combinatorial optimization algorithms over graphs (Dai et al., 2017) and learning branching policies (Khalil et al., 2016). Subsequent efforts have targeted branching decisions (Scavuzzo et al., 2022), primal heuristics (Shen et al., 2021; Angioni et al., 2025), solver parameters (Patel, 2024), and cutting planes (Wang et al., 2023; Dragotto et al., 2023). These methods aim to reach a provably optimal solution faster by improving decisions within the solver.

In this paper we take a different approach: rather than speeding up the search for optimality, we learn when to stop early. We train a neural network to predict the true optimality gap from the solver state and use conformal prediction (Angelopoulos & Bates, 2021) to calibrate a stopping threshold with rigorous probabilistic guarantees on solution

[1]Operations Research and Financial Engineering, Princeton University, Princeton, USA. Correspondence to: Bartolomeo Stellato <bstellato@princeton.edu>.

*Proceedings of the 43ʳᵈ International Conference on Machine Learning*, Seoul, South Korea. PMLR 306, 2026. Copyright 2026 by the author(s).

quality. On benchmarks from the Distributional MIPLIB library (Huang et al., 2024), our method achieves significant speedups while provably returning near-optimal solutions with high probability.

## 2. Related works

### 2.1. Learning to accelerate mixed-integer optimization

Machine learning has emerged as a powerful tool to accelerate mixed-integer optimization (Bengio et al., 2021; Zhang et al., 2022). The most studied application is learning branching policies. Khalil et al. (2016) pioneered imitation learning to approximate strong branching. Gasse et al. (2019) introduced graph neural networks that exploit the bipartite variable-constraint structure of MILPs, inspiring subsequent work on hybrid models combining graph neural networks and multi-layer perceptrons (Gupta et al., 2020) and tree-based formulations (Scavuzzo et al., 2022). Scavuzzo et al. (2024) provide a comprehensive survey of machine learning for branch-and-bound. Beyond branching, researchers have applied learning to cutting plane selection (Wang et al., 2023; Huang et al., 2021; Dragotto et al., 2023), solver configuration (Patel, 2024; Gao et al., 2025; Liu et al., 2024), and primal heuristics (Shen et al., 2021; Nair et al., 2020; Angioni et al., 2025; Han et al., 2023). Some methods learn to predict solutions directly without solving (Sun & Yang, 2023). Apollo-MILP (Liu et al., 2025) uses an alternating prediction-correction loop to fix high-confidence variable values and reduce the problem dimension, using an uncertainty bound to identify reliable predictions; however, it targets primal solution quality rather than certifying when to stop the solver.

All these approaches aim to reach a provably optimal solution faster by improving decisions within the solver. Our method takes a fundamentally different approach: we observe that branch-and-bound often finds an optimal solution long before it can prove optimality (Figure 1), and we learn when to stop early rather than speeding up the proof.

### 2.2. Conformal prediction

Conformal prediction provides distribution-free uncertainty quantification with finite-sample guarantees (Angelopoulos & Bates, 2021). It has been applied across domains including natural language processing (Campos et al., 2024), time-series forecasting (Stankevičiūtė et al., 2021), medical imaging (Vazquez & Facelli, 2022; Fayyad et al., 2023), and autonomous driving (Doula et al., 2024). Recent work has extended conformal prediction to optimization under uncertainty, providing calibration guarantees for robust optimization (Yeh et al., 2025). Clarke & Stellato (2025) used conformal prediction to bound the suboptimality of learned heuristics for parametric MIPs. This work differs

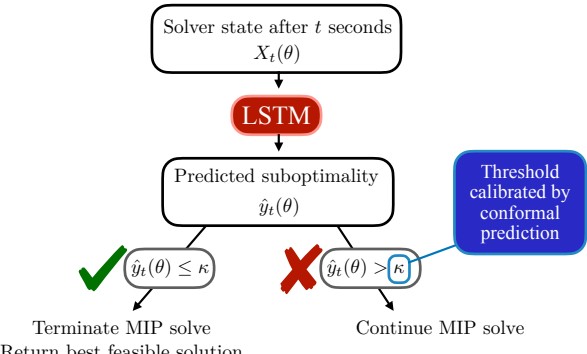

*Figure 2.* Diagram of the proposed conformal prediction method for accelerating MIP solvers by learning to terminate early. The LSTM predicts the suboptimality-gap and the solver terminates and returns the current best solution when the predicted gap is below a threshold. The threshold is chosen by conformal prediction.

from (Clarke & Stellato, 2025) in that we learn a stopping criterion for the exact solver rather than validating an independent heuristic. In the previous work, the learned heuristic and the solution-finding heuristic are independent of one another, and the use of conformal prediction is to give bounds on the error of the learned heuristic. In this work, we explicitly set an acceptable error probability and magnitude of error, and use conformal prediction to calibrate a stopping criterion that achieves these guarantees.

This work is arguably the first to use conformal prediction to accelerate exact MIP solving. Rather than validating an independent heuristic, in this work we learn a termination criterion that stops the solver early while providing probabilistic guarantees on solution quality.

## 3. Parametric mixed integer optimization

This section formalizes the problem setting. We define parametric mixed-integer programs, describe how solvers maintain bounds over time, and show that standard termination criteria are conservative: the solver may hold a near-optimal solution long before it can prove optimality.

### 3.1. Parametric mixed-integer programs

Let $\mathbf{Q}$ be a distribution supported on a set $\Theta \subseteq \mathbf{R}^p$. For each $\theta \in \Theta$ let $\text{MIP}_\theta$ be the optimization problem,

$$z^\star(\theta) = \begin{array}{ll} \text{minimize} & c(\theta)^T x \\ \text{subject to} & x \in \mathcal{X}(\theta), \end{array} \quad (\text{MIP}_\theta)$$

where $\mathcal{X}(\theta) \subseteq \mathbf{R}^m$ is the feasible set, which may include integrality constraints on some variables, and $c(\theta) \in \mathbf{R}^m$ is the cost vector. We consider repeated solves of $(\text{MIP}_\theta)$ for independent draws of $\theta \sim \mathbf{Q}$. We write $x^\star(\theta)$ for an optimal solution to $(\text{MIP}_\theta)$.

### 3.2. Solver behavior over time

Consider a fixed $\theta \in \Theta$. An algorithm $\mathcal{A}(\theta)$ designed to solve ($\mathrm{MIP}_\theta$) can be viewed as a function of time. At time $t \geq 0$ the algorithm holds an upper bound $U_\theta(t)$ and a lower bound $L_\theta(t)$ such that $L_\theta(t) \leq z^\star(\theta) \leq U_\theta(t)$. If $U_\theta(t) < \infty$ then it also holds a feasible solution $x_\theta(t) \in \mathcal{X}(\theta)$ such that $c(\theta)^T x_\theta(t) = U_\theta(t)$. We assume that $\theta$ and $t$ uniquely identify the algorithm state; this approximation holds when problems are solved on the same hardware.

For the solver to eventually certify optimality, the bounds must converge. We formalize this as the following assumption, which holds for any complete branch-and-bound algorithm.

**Assumption 3.1.** $U_\theta(t) - L_\theta(t) \to 0$ as $t \to \infty$.

Define the *true optimality gap* at time $t > 0$ for algorithm $\mathcal{A}(\theta)$ by

$$g_\theta(t) = \frac{U_\theta(t) - z^\star(\theta)}{|z^\star(\theta)|},$$

assuming $z^\star(\theta) \neq 0$. Assumption 3.1 implies that $g_\theta(t) \to 0$ as $t \to \infty$. The true optimality gap is unknown during the solve but can be computed afterward since the solver returns $z^\star(\theta)$.

### 3.3. Standard termination criteria

Solvers typically terminate when the optimality gap

$$g_\theta^{\mathrm{alg}}(t) = \frac{U_\theta(t) - L_\theta(t)}{|L_\theta(t)|},$$

drops below a threshold $\epsilon > 0$. We define the $\epsilon$-optimality stopping time for algorithm $\mathcal{A}$ as

$$\tau_\epsilon(\theta) = \inf\{t \geq 0 \mid g_\theta^{\mathrm{alg}}(t) \leq \epsilon\}.$$

At time $\tau_\epsilon(\theta)$, the best feasible solution $x_\theta(\tau_\epsilon(\theta))$ is $\epsilon$-optimal because $g_\theta(\tau_\epsilon(\theta)) \leq g_\theta^{\mathrm{alg}}(\tau_\epsilon(\theta)) \leq \epsilon$, where the first inequality uses $L_\theta(t) \leq z^\star(\theta)$. However, the first inequality may not be tight. At time $\tau_\epsilon(\theta)$, the algorithm may have held the solution $x_\theta(\tau_\epsilon(\theta))$ for some time already. The algorithm could have terminated earlier and still returned an $\epsilon$-optimal solution. This observation is the foundation of our approach.

### 3.4. Probabilistic termination criteria

At the deterministic stopping time $\tau_\epsilon(\theta)$, we have

$$\mathbf{P}[g_\theta(\tau_\epsilon(\theta)) \leq \epsilon] = 1,$$

where the probability is over $\theta \sim \mathbf{Q}$. We relax this by allowing a small error probability $\alpha \in (0, 1)$, seeking stopping times $\hat{\tau}(\theta)$ satisfying

$$\mathbf{P}[g_\theta(\hat{\tau}(\theta)) \leq \epsilon] \geq 1 - \alpha. \tag{1}$$

Allowing error probability $\alpha$ enables much earlier termination than deterministic criteria. Section 4 describes how we construct such stopping times by learning to approximate $g_\theta(t)$.

## 4. Learning termination criteria

This section develops a learned stopping criterion with rigorous probabilistic guarantees. The key idea is to train a predictor $\hat{g}_\theta(t)$ that approximates the true optimality gap $g_\theta(t)$, then use conformal prediction to calibrate a threshold $\kappa$ such that terminating when $\hat{g}_\theta(t) \leq \kappa$ satisfies (1).

Given predictor $\hat{g}_\theta(t)$, define the stopping time

$$\hat{\tau}_\kappa(\theta) = \inf\{t \geq 0 \mid \hat{g}_\theta(t) \leq \kappa\},$$

where $\kappa$ depends on the desired tolerance $\epsilon$ and error probability $\alpha$. We assume $\hat{g}_\theta(t) \to 0$ as $t \to \infty$ for all $\theta \in \Theta$; Section 4.2 describes how the predictor enforces this property.

### 4.1. Conformal prediction for termination

We develop a conformal prediction framework to choose $\kappa$ given $\epsilon$ and $\alpha$ such that (1) holds. The construction requires two technical definitions.

Define the *left-inverse* of a function $f : \mathbf{R}_+ \to \mathbf{R}_+$ by

$$f^{-1}(x) = \begin{cases} \inf\{t \geq 0 \mid f(t) \leq x\} & x \leq \sup f \\ \infty & \text{otherwise.} \end{cases}$$

Define the *rolling minimum* of the predictor by

$$\bar{g}_\theta(t) = \min_{0 \leq s \leq t} \hat{g}_\theta(s).$$

The function $\bar{g}_\theta(t)$ is nonincreasing in $t$. Let $\hat{g}_\theta^{-1}$ and $g_\theta^{-1}$ denote the left-inverses of $\hat{g}_\theta$ and $g_\theta$. For $k \geq 0$ define $\hat{\tau}_k(\theta) = \hat{g}_\theta^{-1}(k)$. The following theorem, based on conformal prediction theory (Angelopoulos & Bates, 2021), shows how to choose $\kappa$.

**Theorem 4.1.** *Let $c \in \mathbf{Z}_+$ and $n \in \{1, \ldots, c\}$. Let $\theta_1, \ldots, \theta_c, \theta_{c+1} \sim \mathbf{Q}$ be independent. Let*

$$\kappa = \sup\{k \geq 0 \mid \#\{i \in \{1, \ldots, c\} \mid g_{\theta_i}(\hat{\tau}_k(\theta_i)) \leq \epsilon\} \geq n\}.$$

*Then $\mathbf{P}[g_{\theta_{c+1}}(\hat{\tau}_\kappa(\theta_{c+1})) \leq \epsilon] \geq n/(c+1)$.*

The proof relies on the following lemma.

**Lemma 4.2.** *Let $Z_1, \ldots, Z_{c+1}$ be iid random variables in $\mathbf{R}$. Let $Z_{[1]}, \ldots, Z_{[c]}$ denote $Z_1, \ldots, Z_c$ arranged in increasing order. For any $n \in \{1, \ldots, c\}$,*

$$\mathbf{P}[Z_{c+1} \geq Z_{[c+1-n]}] \geq n/(c+1).$$

*Proof.* Let $m$ be the rank of $Z_{c+1}$ among $Z_1, \ldots, Z_{c+1}$ with random tie-breaking. Since $m \geq c + 2 - n$ implies $Z_{c+1} \geq Z_{[c+1-n]}$,

$$\mathbf{P}[Z_{c+1} \geq Z_{[c+1-n]}] \geq \mathbf{P}[m \geq c + 2 - n]$$
$$= \sum_{j=c+2-n}^{c+1} \mathbf{P}[m = j],$$
$$= \sum_{j=c+2-n}^{c+1} \frac{1}{c+1}$$
$$= \frac{n}{c+1}.$$

where the second-last equality holds by exchangeability of $Z_1, \ldots, Z_{c+1}$. $\qquad\square$

*Proof of Theorem 4.1.* By definition, $\kappa$ is the largest $k \geq 0$ such that at least $n$ of $\{g_{\theta_i}(\hat{\tau}_k(\theta_i))\}_{i=1}^c$ are at most $\epsilon$. Since $g_\theta(t)$ is nonincreasing,

$$g_\theta(t) \leq \epsilon \iff g_\theta^{-1}(\epsilon) \leq t.$$

We also have $\bar{g}_\theta^{-1} = \hat{g}_\theta^{-1}$, so

$$\bar{g}_\theta(t) \geq k \iff \bar{g}_\theta^{-1}(k) \geq t \iff \hat{g}_\theta^{-1}(k) \geq t. \quad (2)$$

Observe that,

$$g_{\theta_i}^{-1}(\epsilon) \leq \hat{\tau}_k(\theta_i) \iff g_{\theta_i}^{-1}(\epsilon) \leq \bar{g}_{\theta_i}^{-1}(k)$$
$$\iff \bar{g}_{\theta_i}(g_{\theta_i}^{-1}(\epsilon)) \geq k,$$

where the first equivalence is the definition of $\hat{\tau}_k$ and the second is by (2). Therefore, by definition, $\kappa$ is exactly

$$\sup\{k \geq 0 \mid \#\{i \in \{1, \ldots, c\} \mid \bar{g}_{\theta_i}(g_{\theta_i}^{-1}(\epsilon)) \geq k\} \geq n\},$$

which corresponds to the $n$-th largest element of $\{\bar{g}_{\theta_i}(g_{\theta_i}^{-1}(\epsilon))\}_{i=1}^c$. By Lemma 4.2 applied to $Z_i = \bar{g}_{\theta_i}(g_{\theta_i}^{-1}(\epsilon))$,

$$\mathbf{P}[\bar{g}_{\theta_{c+1}}(g_{\theta_{c+1}}^{-1}(\epsilon)) \geq \kappa] = \mathbf{P}[Z_{c+1} \geq Z_{[c+1-n]}]$$
$$\geq n/(c+1).$$

Reversing the equivalences above yields $\mathbf{P}[g_{\theta_{c+1}}(\hat{g}_{\theta_{c+1}}^{-1}(\kappa)) \leq \epsilon] \geq n/(c+1)$. $\qquad\square$

To apply the theorem, fix tolerance $\epsilon > 0$ and error probability $\alpha > 0$. Given a *calibration dataset* $\mathcal{C} = \{\theta_i\}_{i=1}^c$ of independent draws from $\mathbf{Q}$, choose $n$ such that $n/(c+1) \geq 1 - \alpha$. Theorem 4.1 yields a threshold $\kappa$ such that for any new instance $\theta_{c+1} \sim \mathbf{Q}$ independent of $\mathcal{C}$, terminating at time $\hat{\tau}_\kappa(\theta_{c+1})$ gives

$$\mathbf{P}[g_{\theta_{c+1}}(\hat{\tau}_\kappa(\theta_{c+1})) \leq \epsilon] \geq 1 - \alpha.$$

**Remark (multiple testing).** A naive approach would predict the optimality gap at every solver callback and terminate the first time the prediction drops below $\epsilon$. This constitutes multiple testing: each callback represents a new hypothesis test, and the probability of at least one false positive (stopping before a $\epsilon$-optimal solution has been found) grows with the number of callbacks, which can number in the hundreds per solve. Our conformal approach avoids this issue because the threshold $\kappa$ is calibrated *once* on the calibration set, prior to deployment. The guarantee in Theorem 4.1 covers the *entire* trajectory: the single pre-calibrated threshold $\kappa$ ensures that the first time $\hat{g}_\theta(t) \leq \kappa$ occurs, the solution is $\epsilon$-optimal with the stated probability, with no multiple-testing correction required.

### 4.2. Training the predictor

We parametrize $g_\theta(t)$ using a neural network. Let $X_\theta(t) \in \mathbf{R}^M$ be the solver state vector at time $t$, containing the upper and lower bounds on $z^\star(\theta)$, the current best feasible solution $x_\theta(t)$, the number of explored nodes, the parameter $\theta$, and the elapsed solve time. Let $h : \mathbf{R}^M \to \mathbf{R}_+$ be a neural network. For $l \leq u$ define the squashing function

$$\phi(x \mid l, u) = (u - l)\frac{e^x}{1 + e^x}, \quad (3)$$

which satisfies $\phi(x \mid l, u) \in [0, u - l]$ for all $x \in \mathbf{R}$. The predictor is

$$\hat{g}_\theta(t) = \phi\big(h(X_\theta(t)) \mid L_\theta(t), U_\theta(t)\big).$$

By Assumption 3.1, $U_\theta(t) - L_\theta(t) \to 0$ as $t \to \infty$, so $\hat{g}_\theta(t) \to 0$.

We train the predictor to approximate $U_\theta(t) - z^\star(\theta)$ using the weighted loss

$$\mathcal{L} = \mathbf{E}\left[\int_0^\infty w_\theta(t)\big(\hat{g}_\theta(t) - \big(U_\theta(t) - z^\star(\theta)\big)\big)^2 dt\right],$$

where $w_\theta(t) \geq 0$ is a time-weighting with $\int_0^\infty w_\theta(t)dt = 1$. Let $T_\theta$ denote the termination time of branch-and-bound on problem (MIP$_\theta$). For discretization step $\delta > 0$, let $S_\theta = \lceil T_\theta/\delta \rceil$. The empirical loss over training data $\mathcal{D} = \{\theta_i\}_{i=1}^d$ is

$$\hat{\mathcal{L}} = \frac{1}{d}\sum_{i=1}^d \sum_{t=0}^{S_{\theta_i}} w_{\theta_i}(t\delta)\big(\hat{g}_{\theta_i}(t\delta) - y_{\theta_i}(t\delta)\big)^2, \quad (4)$$

where $y_{\theta_i}(t\delta) = U_{\theta_i}(t\delta) - z^\star(\theta_i)$. We minimize (4) using stochastic gradient descent. The weighting

$$w_\theta(t) \propto \frac{1}{y_\theta(t)} \quad (5)$$

assigns highest weight to times when $y_\theta(t)$ is small. Prediction accuracy matters most near $y_\theta(t) \approx \epsilon$, since this is when the termination decision is made. Because $\epsilon$ is typically small, the weighting (5) focuses the predictor on the critical region where $g_\theta(t) \approx \epsilon$.

### 4.3. Method overview

The overall pipeline comprises three steps.

1. **Training.** Sample $d$ instances $\theta \sim \mathbf{Q}$ to form $\mathcal{D} = \{\theta_i\}_{i=1}^d$ and train the predictor $g_\theta(t)$ by minimizing (4).

2. **Calibration.** Sample $c$ instances $\theta \sim \mathbf{Q}$ to form $\mathcal{C} = \{\theta_i\}_{i=1}^c$. Fix tolerance $\epsilon > 0$ and error probability $\alpha \in (0, 1)$. Compute $\kappa$ via Theorem 4.1.

3. **Deployment.** On a new instance $\theta$, terminate at time $\hat{\tau}_\kappa(\theta)$ (when $g_\theta(t)$ first drops below $\kappa$) and return the best feasible solution.

The returned solution is $\epsilon$-optimal with probability at least $1 - \alpha$ over the calibration set and test instance. Figure 2 illustrates the method.

## 5. Theoretical guarantees

Let the suboptimality at termination be

$$s(\theta, \kappa) = \frac{U_\theta(\hat{\tau}_\kappa(\theta)) - z^\star(\theta)}{|z^\star(\theta)|}.$$

By construction, our method satisfies

$$\mathbf{P}[s(\theta, \kappa) \leq \epsilon] \geq 1 - \alpha, \tag{6}$$

where randomness is over the calibration set $\mathcal{C}$ and $\theta \sim \mathbf{Q}$. Define the empirical and expected suboptimality of threshold $\kappa$ as

$$\hat{S}(\kappa) = \frac{1}{c}\sum_{i=1}^c s(\theta_i, \kappa), \quad S(\kappa) = \mathbf{E}_{\theta \sim \mathbf{Q}}[s(\theta, \kappa) \mid \mathcal{C}],$$

and the empirical and expected runtime as

$$\hat{T}(\kappa) = \frac{1}{c}\sum_{i=1}^c \hat{\tau}_\kappa(\theta_i), \quad T(\kappa) = \mathbf{E}_{\theta \sim \mathbf{Q}}[\hat{\tau}_\kappa(\theta) \mid \mathcal{C}].$$

The guarantee (6) averages over both the calibration set and test point. This section establishes stronger *sample-conditional* guarantees (Duchi, 2025) that hold for a fixed calibration set with high probability:

1. **Expected suboptimality and runtime** (Section 5.1, Theorem 5.1): $S(\kappa) \leq \hat{S}(\kappa) + O(1/\sqrt{c})$ and $T(\kappa) \leq \hat{T}(\kappa) + O(1/\sqrt{c})$, so the expected suboptimality and runtime on new instances are close to their empirical values on calibration data.

2. **Success probability** (Section 5.2, Theorem 5.2): $\mathbf{P}[s(\theta, \kappa) \leq \epsilon \mid \mathcal{C}] \geq 1 - \alpha - O(1/\sqrt{c})$, so the conditional probability of achieving tolerance $\epsilon$ concentrates near $1 - \alpha$.

### 5.1. Expected suboptimality and runtime

We bound the expected suboptimality and runtime conditional on the calibration set. Let $S_{\max}$ bound the suboptimality, *i.e.*, $s(\theta, \kappa) \leq S_{\max}$ for all $\kappa \geq 0$ and $\theta \in \Theta$. Let $T_{\max}$ bound the runtime, *i.e.*, $\hat{\tau}_\kappa(\theta) \leq T_{\max}$ for all $\kappa \geq 0$ and $\theta \in \Theta$. These bounds can be enforced via deterministic stopping criteria.

**Theorem 5.1.** *For any $\delta > 0$, with probability at least $1 - \delta$ over the calibration set $\mathcal{C}$,*

$$S(\kappa) \leq \hat{S}(\kappa) + S_{max}\sqrt{\frac{2\log(ec)}{c}} + S_{max}\sqrt{\frac{\log(1/\delta)}{2c}}.$$

*The runtime satisfies the analogous bound*

$$T(\kappa) \leq \hat{T}(\kappa) + T_{\max}\sqrt{\frac{2\log(ec)}{c}} + T_{\max}\sqrt{\frac{\log(1/\delta)}{2c}}.$$

*Proof of Theorem 5.1.* We prove the suboptimality bound; the runtime bound follows analogously. The proof uses VC-dimension to obtain uniform convergence. For $u \in [0, S_{\max}]$ and $\kappa \in \mathbf{R}_+$ define the classifier

$$\rho(\kappa, u)(\theta) = \mathbf{1}_{s(\theta, \kappa) > u},$$

which equals 1 if $s(\theta, \kappa) > u$ and 0 otherwise. Define the empirical and true risk as

$$\hat{S}(\rho(\kappa, u)) = \frac{1}{c}\sum_{i=1}^c \rho(\kappa, u)(\theta_i),$$

$$S(\rho(\kappa, u)) = \mathbf{E}_{\theta \sim \mathbf{Q}}[\rho(\kappa, u)(\theta)].$$

For fixed $u \in [0, S_{\max}]$ let $\nu(u)$ be the VC-dimension (Definition 3.10, (Mohri et al., 2012)) of the set of functions $\{\rho(\kappa, u) \mid \kappa \in \mathbf{R}_+\}$. Let $\hat{\mathbf{Q}}$ be a uniform distribution on $\mathcal{C}$ (the empirical distribution for the calibration dataset). We have,

$$|S(\kappa) - \hat{S}(\kappa)|$$

$$= \left| \mathbf{E}_{\theta \sim \mathbf{Q}}[s(\theta, \kappa)] - \mathbf{E}_{\theta \sim \hat{\mathbf{Q}}}[s(\theta, \kappa)] \right|$$

$$= \left| \int_0^{S_{\max}} \mathbf{P}_{\theta \sim \mathbf{Q}}[s(\theta, \kappa) > u] - \mathbf{P}_{\theta \sim \hat{\mathbf{Q}}}[s(\theta, \kappa) > u]\mathrm{d}u \right|$$

$$\leq S_{\max} \sup_{u \in [0, S_{\max}]} \left| S(\rho(\kappa, u)) - \hat{S}(\rho(\kappa, u)) \right|$$

$$\leq S_{\max} \sup_{u \in [0, S_{\max}]} \sqrt{\frac{2\nu(u)\log(ec/\nu(u))}{c}} + S_{\max}\sqrt{\frac{\log(1/\delta)}{2c}}.$$

The first equality is the definition of the empirical distribution. The second uses $\mathbf{E}[X] = \int_0^\infty \mathbf{P}[X > u]\, du$ for nonnegative $X$. The third bounds the integral uniformly. The fourth applies Corollary 3.19 of Mohri et al.

(2012). It remains to show $\nu(u) = 1$. The function $\kappa \mapsto \rho(\kappa, u)(\theta)$ is nondecreasing in $\kappa$. For any $\theta_1, \theta_2$, let $\kappa_i = \inf\{\kappa : \rho(\kappa, u)(\theta_i) = 1\}$. If $\kappa_1 \leq \kappa_2$, then $\rho(\kappa, u)(\theta_1) \geq \rho(\kappa, u)(\theta_2)$ for all $\kappa$, so $\{\theta_1, \theta_2\}$ cannot be shattered. Thus $\nu(u) \leq 1$, and since $\nu(u) \geq 1$ by definition, we have $\nu(u) = 1$. $\qquad\square$

## 5.2. Success probability

The probability in (6) is over both the calibration set and test point, which is equivalent to

$$\mathbf{E}_{\mathcal{C}\sim\mathbf{Q}^c}\left[\mathbf{P}_{\theta\sim\mathbf{Q}}[s(\theta,\kappa)\leq\epsilon\mid\mathcal{C}]\right]\geq 1-\alpha.$$

The next result shows that the conditional probability $\mathbf{P}[s(\theta,\kappa)\leq\epsilon\mid\mathcal{C}]$ concentrates near $1-\alpha$. The proof follows from Proposition 2a of Vovk (2012); we include it for completeness.

**Theorem 5.2.** *For any $\delta > 0$, with probability at least $1-\delta$ over the calibration set $\mathcal{C}$,*

$$\mathbf{P}_{\theta\sim\mathbf{Q}}[s(\theta,\kappa)\leq\epsilon\mid\mathcal{C}]\geq 1-\alpha-\sqrt{\frac{\log(2/\delta)}{2c}}.$$

*Proof.* Let $\hat{\mathbf{Q}}$ be the uniform distribution on $\mathcal{C}$ (the distribution over the empirical sample). By definition of $\kappa$ we know that,

$$\mathbf{P}_{\theta\sim\hat{\mathbf{Q}}}[s(\theta,\kappa)\leq\epsilon]\geq 1-\alpha. \qquad (7)$$

Let $\hat{P}_\epsilon = \mathbf{P}_{\theta\sim\hat{\mathbf{Q}}}[s(\theta,\kappa)\leq\epsilon]$ and $P_\epsilon = \mathbf{P}_{\theta\sim\mathbf{Q}}[s(\theta,\kappa)\leq\epsilon]$. By Corollary 1 (Massart, 1990) the following is true, for all $\lambda > 0$, where the randomness is taken over the calibration set $\mathcal{C}$,

$$\mathbf{P}[|\hat{P}_\epsilon - P_\epsilon| \geq \lambda/\sqrt{c}] \leq 2\exp(-2\lambda^2).$$

Set $\delta = 2\exp(-2\lambda^2)$ and rearrange this expression to get,

$$\mathbf{P}\left[|\hat{P}_\epsilon - P_\epsilon| \geq \sqrt{\frac{\log(2/\delta)}{2c}}\right] \leq \delta.$$

This combined with (7) completes the proof of the first inequality. $\qquad\square$

## 5.3. Robustness to distribution shifts

We can also show that the conformal guarantees hold under a shifted distribution $\tilde{\mathbf{Q}}$ which is $\rho$-close to $\mathbf{Q}$ in total variation distance, so the method is robust to distribution shifts.

**Theorem 5.3.** *Let $\tilde{\mathbf{Q}}$ be a distribution such that $\mathrm{TV}(\tilde{\mathbf{Q}}, \mathbf{Q}) \leq \rho$. Then the guarantees of Theorems 5.1 and 5.2 hold with respect to $\tilde{\mathbf{Q}}$, with an additional additive $\rho$ term. In particular, we have,*

$$\mathbf{P}[s(\theta,\kappa)\leq\epsilon]\geq 1-\alpha-\rho,$$

*where the probability is over the shifted distribution test point $\theta \sim \tilde{\mathbf{Q}}$ and calibration set $\mathcal{C} \sim \mathbf{Q}^c$. Further, we have, with probability at least $1-\delta$ over the calibration set $\mathcal{C}$,*

$$\mathbf{P}_{\theta\sim\tilde{\mathbf{Q}}}[s(\theta,\kappa)\leq\epsilon\mid\mathcal{C}]\geq 1-\alpha-\rho-\sqrt{\frac{\log(2/\delta)}{2c}}.$$

*Proof.* Write $\kappa(\mathcal{C})$ for the random variable $\kappa$ as a function of the calibration data $\mathcal{C}$. For the first inequality,

$$\mathbf{P}[s(\theta,\kappa)\leq\epsilon]$$
$$= \mathbf{E}_{\mathcal{C}}\left[\mathbf{P}_{\theta\sim\tilde{\mathbf{Q}}}[s(\theta,\kappa(\mathcal{C}))\leq\epsilon]\right]$$
$$= \mathbf{E}_{\mathcal{C}}\Bigg[\mathbf{P}_{\theta\sim\mathbf{Q}}[s(\theta,\kappa(\mathcal{C}))\leq\epsilon]$$
$$\quad -\left|\mathbf{P}_{\theta\sim\tilde{\mathbf{Q}}}[s(\theta,\kappa(\mathcal{C}))\leq\epsilon] - \mathbf{P}_{\theta\sim\mathbf{Q}}[s(\theta,\kappa(\mathcal{C}))\leq\epsilon]\right|\Bigg]$$
$$\geq \mathbf{E}_{\mathcal{C}}\left[\mathbf{P}_{\theta\sim\mathbf{Q}}[s(\theta,\kappa(\mathcal{C}))\leq\epsilon]\right]-\rho$$
$$\geq 1-\alpha-\rho.$$

For the second inequality let

$$\tilde{P} = \mathbf{P}_{\theta\sim\tilde{\mathbf{Q}}}[\theta\in\{\theta\mid s(\theta,\kappa)\leq\epsilon\}\mid\mathcal{C}]$$

and $P = \mathbf{P}_{\theta\sim\mathbf{Q}}[\theta\in\{\theta\mid s(\theta,\kappa)\leq\epsilon\}\mid\mathcal{C}]$. Then we have,

$$\mathbf{P}_{\theta\sim\tilde{\mathbf{Q}}}[s(\theta,\kappa)\leq\epsilon\mid\mathcal{C}]$$
$$= \mathbf{P}_{\theta\sim\tilde{\mathbf{Q}}}[\theta\in\{\theta\mid s(\theta,\kappa)\leq\epsilon\}\mid\mathcal{C}]$$
$$= \mathbf{P}_{\theta\sim\mathbf{Q}}[\theta\in\{\theta\mid s(\theta,\kappa)\leq\epsilon\}\mid\mathcal{C}]-\left|\tilde{P}-P\right|$$
$$\geq \mathbf{P}_{\theta\sim\mathbf{Q}}[s(\theta,\kappa)\leq\epsilon\mid\mathcal{C}]-\rho$$
$$\geq 1-\alpha-\rho-\sqrt{\frac{\log(2/\delta)}{2c}}.$$
$$\qquad\square$$

## 5.4. Time to termination

The guarantees above hold for any predictor, but meaningful speedup requires an accurate predictor $\hat{g}_\theta(t)$. Taking $\kappa = -\infty$ achieves zero suboptimality but provides no speedup. The threshold $\kappa$ from Theorem 4.1 satisfies all guarantees regardless of predictor quality, but if $\hat{g}_\theta(t)$ is inaccurate, the stopping time $\hat{\tau}_\kappa(\theta)$ may equal the standard termination time $\tau_\epsilon(\theta)$.

# 6. Computational experiments

We test our methods on parametric families of problems from the Distributional MIPLIB library (Huang et al., 2024).

*Table 1.* The number of training, calibration, and test instances for each problem family, along with the corresponding reference. The OTS-medium family uses $c = 28$ and $l = 72$ because only 100 instances are available in total from the real-world dataset.

| Problem | $d$ | $c$ | $l$ | Reference |
|---|---|---|---|---|
| CFLP-medium | 800 | 100 | 100 | Scavuzzo et al. |
| GISP-easy | 800 | 100 | 100 | Hochbaum & Pathria |
| OTS-medium | 800 | 28 | 72 | Huang et al. |
| MMCN-medium-BI | 800 | 100 | 100 | Greening et al. |
| MIS-medium | 800 | 100 | 100 | Huang et al. |
| MVC-medium | 800 | 100 | 100 | Korte & Vygen |

We include a series of plots and figures demonstrating the statistical variety of the datasets considered in Appendix A.

### 6.1. Experiment setup

**Hardware.** All models were trained on a NVIDIA A100 GPU. Each optimization solve took place on AMD EPYC 9334 CPUs, and each optimization solve was allowed access to exactly 1GB of memory, and only one thread.

**Predictor.** We parametrize $h$ by an LSTM with 2 layers and 200 hidden units in each layer, followed by a feedforward neural network with 2 hidden layers and 200 units in each layer. All activation units but the final one are ReLU units. The output dimension is one, and the output represents the approximate true optimality gap at time $t$. We have $\hat{g}_\theta(t) = \phi\big(h(X_\theta(t)) \mid L_\theta(t), U_\theta(t)\big)$. We choose an LSTM because it is a simple model which can detect complex patterns in time-series data, but this is not a necessary part of our work. This learned model can be swapped for the most appropriate model for a given task. In some applications, the optimization problem might be sufficiently hard, or the solve might take sufficiently long, that a simple RNN/LSTM might not perform well. In this case, practitioners may want to swap this component out for a transformer, or include a graph neural network to add awareness of the entire problem formulation.

**Covariates.** At time $t$, the covariate vector $X_\theta(t)$ consists of the current bounds $U_\theta(t)$ and $L_\theta(t)$, their 1, 3, and 5 second rolling averages, the elapsed solve time $t$, and the number of nodes expanded in the branch-and-bound tree. Unlike typical learn-to-optimize settings (Scavuzzo et al., 2022), the predictor does not require access to the full problem formulation, so generating covariate data is efficient.

**Optimization solver.** The solvers we test our method on are Gurobi 12 (Gurobi Optimization, LLC, 2024) and Cardinal Optimization (Ge et al., 2022). We use solver callbacks to evaluate the predictor $\hat{g}_\theta(t)$ as the solve is running. We disable presolve on both solvers, use only a single thread, and set Gurobi MIPFocus parameter to 1. The maximum

allowed solve time is set to 600 seconds. In our experiments we seek to solve all problem instances to within 0.1% accuracy, so we set $\epsilon = 0.001$. We choose $\alpha = 0.05$, so that our solver is likely to return an $\epsilon$-optimal solution at least 95% of the time.

**Evaluation metrics.** For each problem family, we report average suboptimality on test data $\mathcal{T} = \{\theta_i\}_{i=1}^l$,

$$\hat{S} = \frac{1}{l} \sum_{i=1}^l s(\theta_i, \kappa),$$

where $s(\theta_i, \kappa)$ is the relative suboptimality of the solution returned by our method on instance $\theta_i$, as defined in Section 5.1. We also report total solve time and the percent of problems solved to within $\epsilon$-optimality.

**Baselines.** We compare our method to the baseline methods in Table 2. Comparisons to a wider range of baselines are also included in Appendix C.

*Table 2.* Baseline methods.

| Method | Solver | Termination criterion |
|---|---|---|
| GRB | Gurobi | $\epsilon$ optimality |
| GRB1 | Gurobi | 1 solution found |
| GRB3 | Gurobi | 3 solutions found |
| COPT | COPT | $\epsilon$ optimality |
| COPT1 | COPT | 1 solution found |
| COPT3 | COPT | 3 solutions found |

**Data.** We run experiments on capacitated facility location problems (CFLP-medium), generalized independent set problems (GISP-easy), maximum independent set problems (MIS-medium), minimum vertex cover problems (MVC-medium), optimal transmission switching problems (OTS-medium), and middle-mile consolidation network design problems (MMCN-medium-BI). The OTS-medium and MMCN-medium-BI problems come from real-world datasets. These problems are costly to solve, and on some families, Gurobi takes hundreds of seconds to solve most instances. The number of train points $d$, calibration points $c$, and test points $l$ vary depending on the problem family. The details on data and formulations are given in Table 1.

**Code overview.** We first solve all problems in the training dataset $\{\theta_i\}_{i=1}^d$ using solver callbacks and save the solver state $X_t(\theta_i)$ at each point $t$ and true optimality gap $g_{\theta_i}(t)$ at which the callback is called. This process can be computationally intensive. The next step is to train the predictor $\hat{g}_\theta(t)$ using the saved solver states and true optimality gaps. We run the Adam optimizer (Kingma &

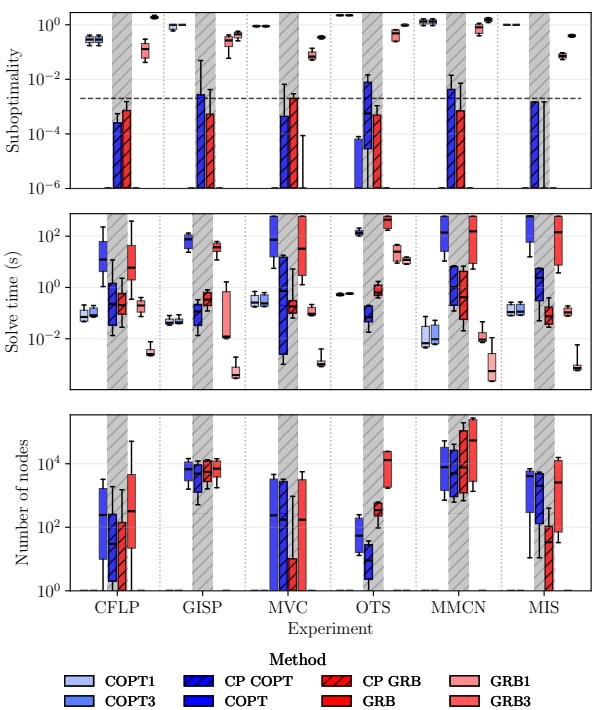

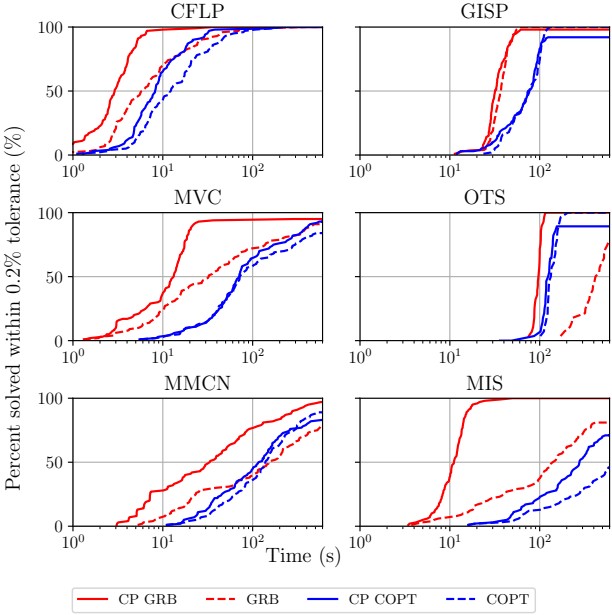

*Figure 4.* The number of instances solved to within $2\epsilon = 0.2\%$-optimality over time for each method on each dataset. Since $\epsilon$ was the conformal target tolerance, we expect almost all problems from each method to be solved to this level of optimality. The conformal prediction method solves most problems very quickly, but it is not guaranteed to solve all problems to within 0.2%-optimality due to the adjustment made by Theorem 4.1, which occurs with probability around $\alpha$.

*Figure 3.* Suboptimality, solve time, and number of nodes explored at termination across all problem families. The dashed line in the top panel marks the target suboptimality $\epsilon = 0.1\%$; values are clipped to $10^{-6}$ when an optimal solution is found. Our methods (shaded grey) achieve substantially faster solve times than the global solvers Gurobi (GRB) and COPT while maintaining low suboptimality.

Ba, 2015) using the PyTorch (Paszke et al., 2019) automatic differentiation software to minimize the loss (4) where $\hat{g}_\theta(t) = \phi\big(h(X_\theta(t)) \mid L_\theta(t), U_\theta(t)\big)$. The training process runs for half an hour on a single GPU for each experiment. We then perform the conformal prediction step described in Section 4.1 to get $\kappa$, and evaluate the quality of the stopping time $\hat{\tau}_\kappa(\theta)$ on the test dataset. Code for our experiments is available through the following link:

    github.com/stellatogrp/conformal_mip

### 6.2. Experiment results

Figure 3 shows suboptimality, solve time, and branch-and-bound nodes at termination for each method. Full tabular results appear in Appendix B. Across all problem families, the conformal prediction method maintains low suboptimality while substantially reducing solve time. Compared to Gurobi, our method terminates 76% faster on CFLP, 78% faster on OTS, 66% faster on MMCN, 88% faster on MVC, and 95% faster on MIS. The improvement on GISP is more modest at 8%, reflecting the structure of these instances: both solvers close the optimality gap relatively quickly after

finding a good solution, leaving less room for early termination.

Figure 4 shows the number of instances solved to $\epsilon$-optimality over time. The conformal prediction method solves at least $100(1-\alpha)\%$ of problems to $\epsilon$-optimality with high probability, though it may not solve all problems due to the probabilistic guarantee of Theorem 4.1. Within the 600-second time limit, our method solves more instances to $\epsilon$-optimality than Gurobi alone on several problem families. The improvement of CP COPT over COPT is more modest but still clear. This difference arises because Gurobi's heuristics find feasible solutions quickly, then spend most of the solve time tightening lower bounds to prove optimality. Our predictor exploits this behavior by estimating when the current solution is already near-optimal, enabling early termination before the lower bound fully closes.

## 7. Conclusion

Branch-and-bound solvers often find optimal solutions long before they can prove optimality. We exploit this by learning a termination criterion that stops the solver early while providing rigorous probabilistic guarantees on solution quality. Our method trains a neural network to predict the true optimality gap from solver state, then uses conformal prediction

to calibrate a stopping threshold that ensures $\epsilon$-optimal solutions with probability at least $1 - \alpha$. We establish sample-conditional guarantees showing that expected suboptimality and success probability concentrate near their empirical values on calibration data. On benchmarks from the Distributional MIPLIB library, our method achieves speedups exceeding 60% on five of six problem families while maintaining 0.1%-optimality with 95% probability.

**Limitations.** Our method requires solving instances from the target distribution during training to record solver state, which can be computationally expensive. This means the approach accelerates problems that current solvers can already solve; it does not extend the frontier of tractable instances. The speedup also varies across problem families: while most benchmarks show improvements exceeding 60%, the GISP family achieves only 8% speedup, suggesting that the gap between finding and proving optimality is problem-dependent. Finally, the conformal guarantee assumes test instances are drawn from the same distribution as calibration data; distribution shift may degrade performance.

## Acknowledgements

Bartolomeo Stellato is supported by the NSF CAREER Award ECCS-2239771 and the ONR YIP Award N000142512147. The authors are pleased to acknowledge that the work reported on in this paper was substantially performed using Princeton University's Research Computing resources.

## Impact Statement

This paper presents work on accelerating combinatorial optimization using probabilistic methods. This has the potential to significantly reduce computational costs in real-world applications where combinatorial optimization problems are prevalent.

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

# A. Experimental data

In this section we display plots and tables to demonstrate that the datasets and convergence curves for the problems considered are nontrivial.

## A.1. Convergence curve analysis

Here we display information on the convergence curves of the MIPs. Convergence curves for 20 instances of each test set are displayed in Figure 5. The following table numerically summarizes the convergence curve data. Here UB conv. (s) is the average time taken (mean $\pm$ SD) for a run of Gurobi to find the optimal solution on the training dataset. LB conv. (s) is the average time taken for a run of Gurobi to reduce the relaxation gap to zero in branch-and-bound. Let $\tau^\star$ be the time at which the optimal solution is discovered by Gurobi. Certified gap at $\tau^\star$ is the relaxation gap defined by the branch-and-bound tree at $\tau^\star$, while Predicted gap at $\tau^\star$ is the gap estimated by our model $g_\theta(\tau^\star)$. Observe that our model reliably predicts that the gap is small at the time when the optimal solution is found. Note that lines in the plot often start some time after zero. This is because lower-bounds only exist once the first node in the branch-and-bound tree has been evaluated and upper-bounds only exist once a feasible solution has been found.

*Table 3.* Convergence statistics across problem families (mean $\pm$ std).

| Family | UB conv. (s) | LB conv. (s) | Certified gap at $\tau^\star$ (%) | Predicted gap at $\tau^\star$ (%) |
|--------|--------------|--------------|-----------------------------------|-----------------------------------|
| CFLP   | $6.8 \pm 7.3$   | $15.6 \pm 40.8$    | $0.108 \pm 0.069$ | $0.034 \pm 0.022$ |
| GISP   | $19.0 \pm 13.0$ | $37.0 \pm 8.9$     | $20.3 \pm 10.3$   | $0.845 \pm 1.791$ |
| MIS    | $10.8 \pm 8.8$  | $221.0 \pm 209.4$  | $1.85 \pm 0.68$   | $0.014 \pm 0.037$ |
| MMCN   | $61.4 \pm 110.7$| $245.2 \pm 230.7$  | $2.67 \pm 1.88$   | $0.149 \pm 0.177$ |
| MVC    | $9.7 \pm 7.8$   | $126.7 \pm 188.1$  | $0.70 \pm 0.18$   | $0.009 \pm 0.020$ |
| OTS    | $84.2 \pm 18.4$ | $431.8 \pm 139.7$  | $4.09 \pm 1.66$   | $0.018 \pm 0.071$ |

## A.2. SCIP Heuristics

We attempt to solve each MIP in each problem test-set with three SCIP (Hojny et al., 2025) heuristics: Feasibility Pump, RENS, and Simple Rounding. We choose the SCIP solver for this task because, unlike the other solvers considered in this paper, it allows the user to enable and disable individual heuristics easily. Results are displayed in Table 4. Individual heuristics do not suffice to reliably solve problems in our dataset.

# B. Tabular results

Time (s) refers to time until termination. Suboptimality refers to the relative suboptimality of the returned solution. Nodes refers to the number of nodes explored in the branch-and-bound tree. Correct refers to the percentage of instances for which the method found an $\epsilon$-optimal solution within the time limit. If the 600 s time limit is exceeded, the solver is allowed to return its current best solution. Speedup (s) refers to the improvement in solve time compared to the baseline method — either CP or GRB depending on the method. Time, Suboptimality, and Nodes are reported as mean $\pm$ standard deviation.

*Table 5.* Capacitated facility location (CFLP-medium).

| Method | Time (s) | Suboptimality | Nodes | Correct | Speedup (s) |
|--------|----------|---------------|-------|---------|-------------|
| CP GRB   | $3.7 \pm 4.2$           | 1.5e-02% $\pm$ 2.8e-04  | $47.9 \pm 169.1$    | 98.0%  | 11.9 |
| GRB      | $15.6 \pm 40.8$         | 0.0% $\pm$ 0            | $1453.5 \pm 5326.4$ | 100.0% | 0    |
| GRB1     | 2.8e-03 $\pm$ 1.0e-03   | 187.7% $\pm$ 9.6e-02    | $0 \pm 0$           | 0.0%   | 15.6 |
| GRB3     | 2.0e-01 $\pm$ 6.2e-02   | 13.0% $\pm$ 4.5e-02     | $1.0 \pm 0$         | 0.0%   | 15.4 |
| CP COPT  | $13.8 \pm 25.2$         | 4.5e-03% $\pm$ 1.1e-04  | $94.0 \pm 258.8$    | 100.0% | 8.2  |
| COPT     | $22.0 \pm 31.1$         | 0.0% $\pm$ 0            | $418.4 \pm 567.5$   | 100.0% | 0    |
| COPT1    | 8.1e-02 $\pm$ 3.0e-02   | 28.8% $\pm$ 4.7e-02     | $1.0 \pm 0$         | 0.0%   | 22.0 |
| COPT3    | 9.6e-02 $\pm$ 2.9e-02   | 28.8% $\pm$ 4.7e-02     | $1.0 \pm 0$         | 0.0%   | 21.9 |

*Table 4.* SCIP heuristic results across problem families.

| Family | Heuristic | Time (s) | Success (%) | Rel. Subopt. (%) | ≤5%-optimal (%) |
|---|---|---|---|---|---|
| CFLP-medium | FP | 7.4 | 100 | 2.64 | 93 |
| | RENS | 3.8 | 100 | 0.09 | 100 |
| | SR | 7.6 | 100 | 12.20 | 3 |
| GISP-easy | FP | 1.0 | 100 | 76.23 | 0 |
| | RENS | 2.2 | 100 | 92.44 | 0 |
| | SR | 1.0 | 100 | 94.92 | 0 |
| MIS-medium | FP | 11.0 | 100 | 122.91 | 0 |
| | RENS | 10.6 | 89 | 100.01 | 0 |
| | SR | 10.7 | 100 | 100.27 | 0 |
| MMCN-medium-BI | FP | 2.1 | 0 | — | — |
| | RENS | 2.5 | 100 | 3.67 | 80 |
| | SR | 2.1 | 2 | 135.63 | 0 |
| MVC-medium | FP | 19.1 | 10 | 37.38 | 0 |
| | RENS | 18.5 | 10 | 89.72 | 0 |
| | SR | 19.3 | 10 | 88.02 | 0 |
| OTS-medium | FP | 188.9 | 36 | 50.36 | 0 |
| | RENS | 200.7 | 0 | — | — |
| | SR | 208.5 | 0 | — | — |

*Table 6.* Generalized independent set (GISP-easy).

| Method | Time (s) | Suboptimality | Nodes | Correct | Speedup (s) |
|---|---|---|---|---|---|
| CP GRB | 34.0 ± 9.4 | 1.1e-02% ± 5.1e-04 | 6265.4 ± 2755.0 | 96.0% | 3.0 |
| GRB | 37.0 ± 8.9 | 0.0% ± 0 | 7332.2 ± 2584.2 | 100.0% | 0 |
| GRB1 | 4.5e-04 ± 2.1e-04 | 44.3% ± 6.6e-02 | 0 ± 0 | 0.0% | 37.0 |
| GRB3 | 1.4e-01 ± 2.9e-01 | 27.3% ± 6.6e-02 | 1.0 ± 0 | 0.0% | 36.9 |
| CP COPT | 67.0 ± 28.4 | 1.7e-01% ± 8.1e-03 | 4771.9 ± 2456.8 | 85.0% | 8.6 |
| COPT | 75.5 ± 27.7 | 0.0% ± 0 | 6672.9 ± 2739.7 | 100.0% | 0 |
| COPT1 | 4.5e-02 ± 8.0e-03 | 96.5% ± 9.9e-02 | 1.0 ± 0 | 0.0% | 75.5 |
| COPT3 | 4.5e-02 ± 7.5e-03 | 100.0% ± 0 | 1.0 ± 0 | 0.0% | 75.5 |

*Table 7.* Optimal transmission switching (OTS-medium).

| Method | Time (s) | Suboptimality | Nodes | Correct | Speedup (s) |
|---|---|---|---|---|---|
| CP GRB | 96.2 ± 9.2 | 7.4e-03% ± 2.3e-04 | 368.3 ± 108.9 | 96.4% | 335.6 |
| GRB | 431.8 ± 139.7 | 3.1e-06% ± 1.6e-07 | 13254.9 ± 6916.7 | 100.0% | 0 |
| GRB1 | 11.5 ± 1.8 | 97.7% ± 3.7e-02 | 1.0 ± 0 | 0.0% | 420.3 |
| GRB3 | 26.2 ± 14.2 | 48.6% ± 1.3e-01 | 1.0 ± 0 | 0.0% | 405.6 |
| CP COPT | 117.8 ± 25.8 | 1.5e-01% ± 3.2e-03 | 12.0 ± 8.9 | 78.6% | 20.7 |
| COPT | 138.5 ± 22.0 | 1.3e-03% ± 2.4e-05 | 67.4 ± 56.6 | 100.0% | 0 |
| COPT1 | 5.2e-01 ± 3.0e-02 | 221.5% ± 9.1e-03 | 1.0 ± 0 | 0.0% | 138.0 |
| COPT3 | 5.8e-01 ± 1.4e-02 | 221.5% ± 9.1e-03 | 1.0 ± 0 | 0.0% | 137.9 |

*Table 8.* Middle-mile consolidation network design (MMCN-medium-BI).

| Method | Time (s) | Suboptimality | Nodes | Correct | Speedup (s) |
|--------|----------|---------------|-------|---------|-------------|
| CP GRB | 83.9 ± 124.9 | 1.8e-02% ± 9.5e-04 | 26635.8 ± 40107.9 | 96.0% | 161.3 |
| GRB | 245.2 ± 230.7 | 0.0% ± 0 | 88856.8 ± 86889.5 | 100.0% | 0 |
| GRB1 | 9.4e-04 ± 1.4e-03 | 153.3% ± 1.2e-01 | 0 ± 0 | 0.0% | 245.2 |
| GRB3 | 1.0e-02 ± 5.1e-03 | 77.3% ± 1.9e-01 | 1.0 ± 0 | 0.0% | 245.2 |
| CP COPT | 148.6 ± 154.8 | 7.6e-02% ± 1.9e-03 | 7939.3 ± 8294.9 | 80.0% | 55.4 |
| COPT | 204.0 ± 179.6 | 0.0% ± 0 | 11426.8 ± 11073.7 | 100.0% | 0 |
| COPT1 | 1.2e-02 ± 1.2e-02 | 130.5% ± 1.5e-01 | 1.0 ± 0 | 0.0% | 204.0 |
| COPT3 | 1.4e-02 ± 9.4e-03 | 130.5% ± 1.5e-01 | 1.0 ± 0 | 0.0% | 204.0 |

*Table 9.* Maximum independent set (MIS-medium).

| Method | Time (s) | Suboptimality | Nodes | Correct | Speedup (s) |
|--------|----------|---------------|-------|---------|-------------|
| CP GRB | 11.7 ± 5.9 | 4.4e-03% ± 2.5e-04 | 46.7 ± 54.3 | 97.0% | 209.2 |
| GRB | 221.0 ± 209.4 | 0.0% ± 0 | 3994.9 ± 3799.9 | 100.0% | 0 |
| GRB1 | 7.9e-04 ± 5.1e-04 | 39.8% ± 1.9e-02 | 0 ± 0 | 0.0% | 221.0 |
| GRB3 | 1.1e-01 ± 2.4e-02 | 7.3% ± 8.7e-03 | 1.0 ± 0 | 0.0% | 220.9 |
| CP COPT | 319.6 ± 212.8 | 1.3e-02% ± 4.2e-04 | 2323.3 ± 1638.4 | 91.0% | 123.8 |
| COPT | 443.4 ± 208.4 | 0.0% ± 0 | 3542.8 ± 1752.2 | 100.0% | 0 |
| COPT1 | 1.2e-01 ± 4.5e-02 | 100.0% ± 0 | 1.0 ± 0 | 0.0% | 443.3 |
| COPT3 | 1.3e-01 ± 4.5e-02 | 100.0% ± 0 | 1.0 ± 0 | 0.0% | 443.3 |

*Table 10.* Minimum vertex cover (MVC-medium).

| Method | Time (s) | Suboptimality | Nodes | Correct | Speedup (s) |
|--------|----------|---------------|-------|---------|-------------|
| CP GRB | 14.3 ± 28.8 | 3.2e-02% ± 6.7e-04 | 12.2 ± 93.6 | 88.0% | 112.4 |
| GRB | 126.7 ± 188.1 | 8.7e-05% ± 8.6e-06 | 727.7 ± 1092.4 | 100.0% | 0 |
| GRB1 | 1.1e-03 ± 3.3e-04 | 34.9% ± 1.9e-02 | 0 ± 0 | 0.0% | 126.7 |
| GRB3 | 1.0e-01 ± 2.5e-02 | 7.4% ± 1.6e-02 | 1.0 ± 0 | 0.0% | 126.6 |
| CP COPT | 145.2 ± 166.4 | 1.1e-02% ± 6.7e-04 | 602.2 ± 888.7 | 98.0% | 44.7 |
| COPT | 190.0 ± 209.7 | 0.0% ± 0 | 869.9 ± 1163.1 | 100.0% | 0 |
| COPT1 | 3.0e-01 ± 1.1e-01 | 89.5% ± 1.9e-02 | 1.0 ± 0 | 0.0% | 189.7 |
| COPT3 | 3.0e-01 ± 1.2e-01 | 89.5% ± 1.9e-02 | 1.0 ± 0 | 0.0% | 189.7 |

## C. Further simple baseline experiments

To further demonstrate that our method learns nontrivial patterns in the convergence curves, we provide comparisons between our methods and some simple baseline models. We consider two further simple models. In the first (linear) the LSTM is replaced by a generalized linear model which is trained to predict $\log(\hat{g}_\theta(t))$ from $X_\theta(t)$, and the conformal prediction step is performed as normal. In the second (CP UB rate) we replace the LSTM predictor with a heuristic which outputs the change in upper bound over the last 5 seconds. The conformal prediction is performed as normal after this. We further compare our method to Gurobi with a larger range of tolerances, given by 1%, 2%, 5%, and 10% (0.1% is given in the tables in Appendix B).

### C.1. Heuristic models

Tabular results for the heuristic models are displayed in Table 11 (relative suboptimality) and Table 12 (solve time).

### C.2. Additional solver tolerances

Results for a wider range of Gurobi solver tolerances (MIPGap) are displayed in Table 13 (relative suboptimality) and Table 14 (solve time). Tolerances of 1% and 2% usually terminate after the optimal solution has been found, but on some problems they take significantly longer than our method and provide no statistical guarantees on optimality.

*Table 11.* Suboptimality (%) by family for LSTM (ours) and the described heuristic models (mean ± std).

| Family | LSTM (ours) | Linear | CP UB rate | Full solve |
|---|---|---|---|---|
| CFLP | 0.018 ± 0.037 | 0.003 ± 0.007 | 0.013 ± 0.046 | 0.003 ± 0.007 |
| GISP | 0.002 ± 0.012 | 0.000 ± 0.000 | 0.000 ± 0.000 | 0.000 ± 0.000 |
| MIS | 0.006 ± 0.029 | 0.000 ± 0.000 | 0.000 ± 0.000 | 0.000 ± 0.000 |
| MMCN | 0.032 ± 0.111 | 0.000 ± 0.000 | 0.000 ± 0.000 | 0.000 ± 0.000 |
| MVC | 0.047 ± 0.076 | 0.002 ± 0.010 | 0.017 ± 0.063 | 0.002 ± 0.010 |
| OTS | 0.001 ± 0.002 | 0.076 ± 0.207 | 0.076 ± 0.207 | 0.000 ± 0.000 |

*Table 12.* Solve time by family for LSTM (ours) and the described heuristic models (mean ± std).

| Family | LSTM (ours) | Linear | CP UB rate | Full solve |
|---|---|---|---|---|
| CFLP | 3.7 ± 4.2 | 7.338 ± 22.543 | 2.949 ± 1.655 | 15.6 ± 40.8 |
| GISP | 34.0 ± 9.4 | 31.602 ± 7.404 | 31.581 ± 7.406 | 37.0 ± 8.9 |
| MIS | 11.7 ± 5.9 | 207.298 ± 207.103 | 208.000 ± 207.179 | 221.0 ± 209.4 |
| MMCN | 83.9 ± 124.9 | 237.573 ± 228.088 | 237.590 ± 228.117 | 245.2 ± 230.7 |
| MVC | 14.3 ± 28.8 | 116.111 ± 179.861 | 10.042 ± 4.495 | 126.7 ± 188.1 |
| OTS | 96.2 ± 9.2 | 245.919 ± 175.426 | 246.270 ± 175.545 | 431.8 ± 139.7 |

## D. Sensitivity analyses

In this section we perform two additional experiments. In the first we consider perturbations of training data to investigate how quickly performance degrades out of distribution. In the second we investigate the effect of changing the size of the calibration set on coverage rate. We only consider the parametric families CFLP, GISP, MIS and MVC in this section because the experiments require repeated resampling of iid datapoints. This is not possible on the datasets which come from real data.

### D.1. Sensitivity to distribution shifts

We used distributional MIPLIB generators for four families to investigate the effect of distribution shift. On each problem which comes with a generator (CFLP, GISP, MIS and MVC), we generate a new train and calibration dataset, of size 1000 and 100 respectively, and generate 100 separate iid test sets of size 100. In each experiment we perturb all random continuous parameters in the generator of the test dataset by multiplying by a uniform random variable $U[1, 1 + p]$, where $p$ controls the perturbation magnitude. We perform the conformal procedure as described in the main text. Results are displayed in Table 15. We observe no systematic degradation in suboptimality, consistent with the TV-distance guarantee.

### D.2. Sensitivity to calibration set size

On each problem which comes with a generator (CFLP, GISP, MIS and MVC), we generate a new train and test dataset, of size 1000 and 100 respectively, and generate 100 separate iid calibration sets of size 100. The conformal procedure is carried out as described in the main paper, and results are reported on the test dataset. Results are displayed in Table 16. Note that because we have regenerated the problems ourselves, to allow repeated resampling of the data, the solve-times do not always line up with the solve-times of problems from the Distributional MIPLIB dataset. For some problems, we were unable to figure out exactly which parameters were used to generate the data. However, the results tell the expected story. Variance of coverage is very high when small calibration sets are used, and it decreases as the size of the calibration set increases. Solve time increases on average as the calibration set size increases, but coverage is more reliable.

*Table 13.* Suboptimality (%) by family for LSTM (ours) and Gurobi at various tolerances (mean ± std).

| Family | LSTM (ours) | 1% | 2% | 5% | 10% |
|---|---|---|---|---|---|
| CFLP | 1.5e-02 ± 2.8e-04 | 0.408 ± 0.277 | 0.918 ± 0.606 | 2.069 ± 1.266 | 6.427 ± 3.634 |
| GISP | 1.1e-02 ± 5.1e-04 | 0.000 ± 0.000 | 0.000 ± 0.000 | 0.046 ± 0.458 | 0.102 ± 0.645 |
| MIS | 4.4e-03 ± 2.5e-04 | 0.014 ± 0.066 | 0.178 ± 0.264 | 1.436 ± 0.521 | 3.288 ± 1.510 |
| MMCN | 1.8e-02 ± 9.5e-04 | 0.003 ± 0.018 | 0.029 ± 0.094 | 0.608 ± 0.555 | 1.544 ± 0.945 |
| MVC | 3.2e-02 ± 6.7e-04 | 0.167 ± 0.139 | 0.484 ± 0.214 | 1.997 ± 0.986 | 5.362 ± 1.018 |
| OTS | 7.4e-03 ± 2.3e-04 | 0.000 ± 0.002 | 0.000 ± 0.002 | 0.075 ± 0.110 | 2.634 ± 1.968 |

*Table 14.* Solve time by family for LSTM (ours) and Gurobi at various tolerances (mean ± std).

| Family | LSTM (ours) | 1% | 2% | 5% | 10% |
|---|---|---|---|---|---|
| CFLP | 3.7 ± 4.2 | 1.724 ± 1.611 | 0.732 ± 0.584 | 0.413 ± 0.135 | 0.354 ± 0.135 |
| GISP | 34.0 ± 9.4 | 44.933 ± 13.197 | 44.182 ± 12.505 | 44.072 ± 13.291 | 39.734 ± 12.164 |
| MIS | 11.7 ± 5.9 | 144.374 ± 168.723 | 33.242 ± 68.120 | 1.606 ± 0.796 | 1.067 ± 0.290 |
| MMCN | 83.9 ± 124.9 | 213.320 ± 218.386 | 105.692 ± 176.299 | 6.854 ± 17.762 | 2.149 ± 1.048 |
| MVC | 14.3 ± 28.8 | 33.798 ± 94.832 | 6.190 ± 20.474 | 2.609 ± 1.002 | 0.445 ± 0.533 |
| OTS | 96.2 ± 9.2 | 74.212 ± 8.474 | 73.323 ± 7.718 | 54.546 ± 8.845 | 41.360 ± 5.283 |

*Table 15.* Suboptimality (%) under distribution shift (mean ± std).

| Family | $p = 0$ | $p = 0.01$ | $p = 0.05$ | $p = 0.2$ |
|---|---|---|---|---|
| CFLP | 0.020 ± 0.059 | 0.023 ± 0.046 | 0.009 ± 0.027 | 0.010 ± 0.031 |
| GISP | 0.055 ± 0.462 | 0.135 ± 0.749 | 0.091 ± 0.574 | 0.075 ± 0.327 |
| MIS | 0.007 ± 0.032 | 0.006 ± 0.026 | 0.012 ± 0.064 | 0.004 ± 0.014 |
| MVC | 0.031 ± 0.071 | 0.016 ± 0.043 | 0.013 ± 0.033 | 0.024 ± 0.051 |

*Table 16.* Effect of the calibration set size $c$ on coverage and solve time.

*(a)* Coverage rate (%).

| Family | $c = 5$ | $c = 10$ | $c = 20$ | $c = 100$ |
|---|---|---|---|---|
| CFLP | 80.0 ± 16.4 | 87.9 ± 9.1 | 88.8 ± 8.6 | 92.7 ± 2.7 |
| GISP | 75.1 ± 18.2 | 80.5 ± 12.0 | 89.7 ± 7.4 | 92.8 ± 3.3 |
| MIS | 77.5 ± 12.5 | 86.8 ± 5.6 | 91.4 ± 5.5 | 95.3 ± 3.0 |
| MVC | 77.7 ± 15.4 | 85.3 ± 11.3 | 90.5 ± 7.5 | 96.0 ± 1.5 |

*(b)* Mean solve time (s).

| Family | $c = 5$ | $c = 10$ | $c = 20$ | $c = 100$ |
|---|---|---|---|---|
| CFLP | 9.4 ± 4.6 | 11.3 ± 3.2 | 11.4 ± 3.0 | 13.1 ± 1.6 |
| GISP | 20.7 ± 7.0 | 22.0 ± 5.6 | 26.8 ± 3.9 | 28.1 ± 2.0 |
| MIS | 8.7 ± 1.1 | 10.1 ± 2.3 | 15.1 ± 13.1 | 14.7 ± 4.3 |
| MVC | 8.8 ± 0.9 | 9.3 ± 0.6 | 9.6 ± 0.3 | 9.7 ± 0.0 |

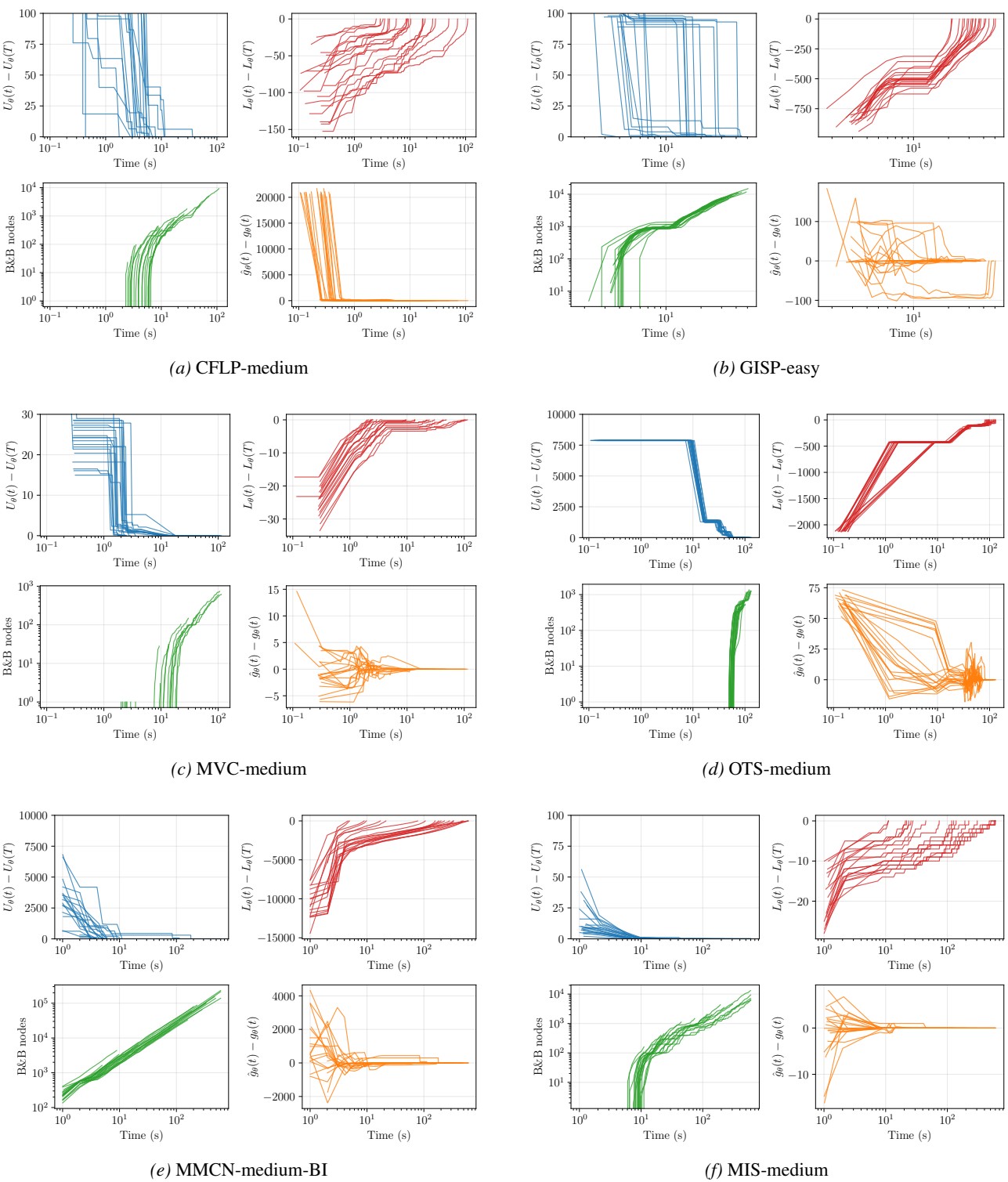

*(a)* CFLP-medium

*(b)* GISP-easy

*(c)* MVC-medium

*(d)* OTS-medium

*(e)* MMCN-medium-BI

*(f)* MIS-medium

*Figure 5.* Convergence curves for 20 test instances from each problem family.

