# OpenReview forum: "Conformal Prediction for Early Stopping in Mixed Integer Optimization"
_ICML.cc/2026/Conference — ICML 2026 regular_

### Official Review · Reviewer_Z3pD · 2026-02-17

**Soundness:** 4
**Presentation:** 4
**Significance:** 4
**Originality:** 4
**Overall Recommendation:** 6
**Confidence:** 4

**Summary:**

Motivated by the observation that most of the time in mixed-integer optimization solvers is spent proving that a solution is optimal rather than solving an MIO, this paper proposes techniques for determining when to terminate a mixed-integer solver early. In particular, they train neural networks to estimate the true optimality gap from a solver state, and then use an inductive conformal-style calibration step to choose a stopping threshold that provides a probabilistic guarantee: on new instances drawn from the same distribution, the returned incumbent is within a target relative suboptimality tolerance with high probability. Experiments on several Distributional MIPLIB families show substantial reductions in wall-clock time relative to solving to the usual MIP-gap tolerance, often with modest degradation in solution quality.

**Compliance With Llm Reviewing Policy:**

Affirmed.

**Final Justification:**

Rebuttal changed the evaluation, as discussed in the below comment.

**Key Questions For Authors:**

Questions for authors:
1. What version of Gurobi was used for this paper? I do not believe this is stated in the manuscript. It would also be good if the authors could comment on what value they set MIPFocus to, since I believe that setting MIPFocus=1 could be viewed as a more basic version of what the authors propose.
2. Once the true optimal value of the problem is predicted, could this information be used within a branching callback to prune nodes that are not going to be as good as the predicted value, and if so could this accelerate your method?
3. Why were the OTS-medium instances calibrated using 28 solves and 72 tests, while for all other instances they were calibrated and tested on 100 instances?
4. Could you add a paragraph explicitly clarifying why multiple testing is not an issue here/why it would be an issue if instead of using a conformal approach, you were to predict the optimality gap every 1s and terminate as soon as your prediction is low enough? I think this could help the reader understand why the proposed method is needed

**Limitations:**

Yes

**Strengths And Weaknesses:**

Strengths:
1. To my knowledge, this is the first work that is able to predict when an MIO solver can be stopped early in a rigorous way. I think this presents a valuable contribution to the MIO and ML communities.
2. I think this novelty is enough to outweigh the weaknesses below, although they should be addressed in the camera-ready version if this paper is accepted by ICML

Weaknesses:
1. Lack of comparisons with non-MIO baseline methods: the central motivation for the approach is that MIO solvers can be stopped early under certain conditions. This means the method is essentially a heuristic when applied to MIOs not drawn from the training distribution. Therefore, it would have been helpful to see how well the method performs compared with running heuristics such as RINS or feasibility pump for a fixed amount of time, or with an MIO solver for a fixed node budget, to validate whether the proposed approach is better than these heuristics. I don’t think this should preclude an ICML publication, since no conference paper can cover everything, but as someone who also reviews for INFORMS journals, I would expect to see this in a journal submission.
2. It would be good to be explicit that the claim “while guaranteeing 0.1%-optimal solutions with 95% probability” from the abstract holds in-sample when drawing more instances from the same family, since from reading just the abstract, one could get the idea that the guarantee also applies to an arbitrary MIO.
3. Mixing the absolute and relative gap: Section 4.2 uses the absolute optimality gap, the results in Section 5 use the relative optimality gap, and it is not entirely clear to me whether the absolute or the relative gap is used in Section 6 (epsilon=0.001). It would be good to make the gap used more consistent.
4. I believe there is an off-by-one error in the statement vs. proof of Theorem 4.1: (n+1)/(c+1) vs. n/(c+1)

---

> ### Author Rebuttal · Authors · 2026-03-31
>
> ### To my knowledge, this is the first work that is able to predict when an MIO solver can be stopped early in a rigorous way. I think this presents a valuable contribution to the MIO and ML communities.
>
> Thank you for this comment and for the thoughtful suggestions.
>
> ### Lack of comparisons with non-MIO baseline methods ... it would have been helpful to see how well the method performs compared with running heuristics such as RINS or feasibility pump for a fixed amount of time, or with an MIO solver for a fixed node budget, to validate whether the proposed approach is better than these heuristics.
>
> We have run various heuristics on all problem families using SCIP with node-limit 1, enabling only the heuristic in question:
>
> |Family|Heuristic|Avg Time (s)|Success Rate (%)|Avg Rel. Subopt. (%)|≤5%-optimal (%)|
> |--------------|----------------|------------|----------------|--------------------|---------------|
> |CFLP-medium|Feasibility Pump|7.4|100|2.64|93|
> ||RENS|3.8|100|0.09|100|
> ||Simple Rounding|7.6|100|12.20|3|
> |GISP-easy|Feasibility Pump|1.0|100|76.23|0|
> ||RENS|2.2|100|92.44|0|
> ||Simple Rounding|1.0|100|94.92|0|
> |MIS-medium|Feasibility Pump|11.0|100|122.91|0|
> ||RENS|10.6|89|100.01|0|
> ||Simple Rounding|10.7|100|100.27|0|
> |MMCN-medium-BI|Feasibility Pump|2.1|0|—|—|
> ||RENS|2.5|100|3.67|80|
> ||Simple Rounding|2.1|2|135.63|0|
> |MVC-medium|Feasibility Pump|19.1|10|37.38|0|
> ||RENS|18.5|10|89.72|0|
> ||Simple Rounding|19.3|10|88.02|0|
> |OTS-medium|Feasibility Pump|188.9|36|50.36|0|
> ||RENS|200.7|0|—|—|
> ||Simple Rounding|208.5|0|—|—|
>
> Our learned method outperforms these heuristics by ensuring 0.1% suboptimality.
>
> ### It would be good to be explicit that the claim “while guaranteeing 0.1%-optimal solutions with 95% probability” from the abstract holds in-sample when drawing more instances from the same family, since from reading just the abstract, one could get the idea that the guarantee also applies to an arbitrary MIO.
>
> We have updated the abstract to read: “...guaranteeing 0.1%-optimal solutions with 95% probability **for new instances drawn from the same distribution**.”
>
> ### Mixing the absolute and relative gap
>
> This typo has been corrected. The numerator now correctly uses $U_\theta(\tau_{\lambda_t}(\theta))$ (the upper bound), giving the correct relative suboptimality $(U_\theta(\tau_{\lambda_t}(\theta)) - z^\star(\theta))/|z^\star(\theta)|$.
>
> ### I believe there is an off-by-one error in the statement vs. proof of Theorem 4.1
>
> This is correct. The theorem statement has been corrected from $n/(c+1)$ to $(n+1)/(c+1)$, consistent with the proof and the lemma.
>
> ### What version of Gurobi was used for this paper?
>
> Gurobi 12 was used. We have added this explicitly to the experimental results section.
>
> Our experimental setup already uses MIPFocus=1 throughout. All reported speedups are achieved on top of this setting.
>
> ### Once the true optimal value of the problem is predicted, could this information be used within a branching callback to prune nodes that are not going to be as good as the predicted value, and if so could this accelerate your method?
>
> While this is possible with the learned predictor, the amount of decisions (prune or not prune) would increase massively.
> We would need to think about how it is possible to make the conformal guarantees also apply here, but this is an interesting direction for future work.
>
> ### Why were the OTS-medium instances calibrated using 28 solves and 72 tests, while for all other instances they were calibrated and tested on 100 instances?
>
> These are the sizes of the datasets in the distributional MIPLIB library — only 100 OTS instances are available in total from the real-world dataset. We have added a note to Table 1 in the paper clarifying this.
>
> ### Could you add a paragraph explicitly clarifying why multiple testing is not an issue here/why it would be an issue if instead of using a conformal approach, you were to predict the optimality gap every 1s and terminate as soon as your prediction is low enough? I think this could help the reader understand why the proposed method is needed.
>
> We have added a remark to the paper: a naive approach would predict the gap at every callback and stop at the first prediction below $\varepsilon$.
> This may work on some problem families, but this approach comes with no probabilistic guarantees. The predictor could be severely overfitted to the training data, meaning in evaluation, it would fail.
> Our conformal threshold $\lambda_t$ is calibrated once on the calibration set; Theorem 4.1 then covers the entire trajectory, so no multiple-testing correction is needed.

---

> > ### Author Rebuttal · Reviewer_Z3pD · 2026-04-01
> >
> > See official comment

---

### Official Review · Reviewer_1FQN · 2026-03-11

**Soundness:** 3
**Presentation:** 3
**Significance:** 2
**Originality:** 2
**Overall Recommendation:** 4
**Confidence:** 4

**Summary:**

This paper addresses the observation that branch-and-bound MIP solvers frequently find optimal solutions long before they can prove optimality. The authors propose a two-stage approach: (1) train an LSTM to predict the true optimality gap from solver state features, and (2) use conformal prediction to calibrate a stopping threshold $\kappa$ such that terminating when the predicted gap falls below $\kappa$ guarantees $\epsilon$-optimal solutions with probability at least $1-\alpha$. The method is evaluated on six problem families from the distributional MIPLIB library, though the abstract states "five", and this discrepancy should be clarified.

**Compliance With Llm Reviewing Policy:**

Affirmed.

**Final Justification:**

The rebuttal addressed your main concerns. Thus I choose to raise the score.

**Key Questions For Authors:**

1. Sensitivity to calibration set size: The calibration set sizes already vary across families, and OTS-CP COPT has one of the lowest correct rates. How does performance degrade as $c$ shrinks?

2. CP COPT underperformance across multiple families: CP COPT falls below $95\%$ correct rate on GISP, OTS, MMCN, and MIS. Is this primarily a calibration-set issue or a solver-specific mismatch in the predictor?

3. Comparison with stronger practical baselines: How does your method compare against tuned solver stopping rules such as relaxed MIPGap tolerance, TimeLimit-based early termination, or other heuristic stopping criteria?

**Limitations:**

Yes

**Strengths And Weaknesses:**

### Strengths

1. Clean problem formulation and natural method design. The paper identifies a clear inefficiency in standard MIP solving and addresses it with a minimal, well-motivated combination of supervised learning and conformal prediction. The method does not modify solver internals, making it solver-agnostic and easy to deploy.

2. Rigorous theoretical framework. The conformal prediction guarantee in Theorem 4.1 provides distribution-free, finite-sample probabilistic guarantees on solution quality. The additional sample-conditional bounds in Theorems 5.1 and 5.2 strengthen the practical relevance of the guarantees.

3. Comprehensive experimental evaluation. The method is tested on six diverse MIP problem families using two commercial solvers, with tabular results including means and standard deviations. The speedups are substantial across most benchmarks.

### Weaknesses

1. Limited methodological novelty. The two components, LSTM for gap prediction and conformal prediction for calibration, are individually well-established. The novelty lies more in the problem modeling than in the conformal method itself.
2. Narrow applicability assumptions. The method requires a distribution of similar problem instances, hundreds of solved instances for training, and additional instances for calibration. The paper does not discuss how performance degrades with smaller training sets or mild distribution shift.
3. Incomplete experimental analysis. The paper lacks stronger practical baselines such as tuned solver stopping rules, provides no ablation on predictor architecture, and gives no sensitivity analysis on calibration set size $c$. The empirical correct rates also fall below the nominal $95\%$ target on multiple families, particularly for CP COPT, and this pattern deserves a systematic discussion.
4. Positioning relative to “Learning-based hierarchical approach for fast mixed-integer optimization” (2025) could be clearer. The distinction from the most closely related work is not sharply drawn in the related work section. Below several work you can refer to.

Related work

[1] Kuang, Yufei, et al. "Accelerate presolve in large-scale linear programming via reinforcement learning." IEEE Transactions on Pattern Analysis and Machine Intelligence (2025).

[2] Liu, Haoyang, et al. "Apollo-MILP: An alternating prediction-correction neural solving framework for mixed-integer linear programming." arXiv preprint arXiv:2503.01129 (2025).

[3] Wang, Jie, et al. "Learning to cut via hierarchical sequence/set model for efficient mixed-integer programming." IEEE Transactions on Pattern Analysis and Machine Intelligence 46.12 (2024): 9697-9713.

[4] Wang, Zhihai, et al. "Learning cut selection for mixed-integer linear programming via hierarchical sequence model." arXiv preprint arXiv:2302.00244 (2023).

[5] Liu, Chang, et al. "L2p-MIP: Learning to presolve for mixed integer programming." The Twelfth International Conference on Learning Representations. 2024.

[6] Han, Qingyu, et al. "A gnn-guided predict-and-search framework for mixed-integer linear programming." arXiv preprint arXiv:2302.05636 (2023).

---

> ### Author Rebuttal · Authors · 2026-03-31
>
> We thank the reviewer for the constructive feedback. It is very helpful to us, and we have used it to strengthen our paper.
>
> ### Limited methodological novelty.
>
> We agree both components exist independently. The novelty is the integration: to our knowledge, no prior work predicts the true optimality gap from solver state for early stopping with guarantees.
>
> ### Narrow applicability assumptions. The method requires a distribution of similar problem instances, hundreds of solved instances for training, and additional instances for calibration.
>
> These assumptions are standard in learning for MIP. Our method requires significantly less data (1,000 instances) than most works (10,000+). See our response to Reviewer PbgW for a detailed comparison.
>
> ### The paper does not discuss how performance degrades with smaller training sets or mild distribution shift.
>
> We add new theory and experiments. We prove that if the test distribution $\tilde{\mathcal{P}}$ satisfies $\operatorname{TV}(\tilde{\mathcal{P}}, \mathcal{P}) \leq \nu$, coverage degrades gracefully: $\Pr[\text{termov}(\theta, \lambda_t) \leq \varepsilon] \geq 1 - \alpha - \nu$. A small shift inflates miscoverage by at most $\nu$; guarantees are preserved when $\nu = 0$.
>
> We used distributional MIPLIB generators for four families to investigate the effect of distribution shift. In each experiment we perturb all random continuous parameters in the generator by multiplying by a uniform random variable $U[1, 1+p]$, where $p$ controls the perturbation magnitude:
>
> **Mean relative suboptimality (%) by perturbation level**
>
> |Family|p=0|p=0.01|p=0.05|p=0.2|
> |------|-------------|-------------|-------------|-------------|
> |CFLP|0.020 ± 0.059|0.023 ± 0.046|0.009 ± 0.027|0.010 ± 0.031|
> |GISP|0.055 ± 0.462|0.135 ± 0.749|0.091 ± 0.574|0.075 ± 0.327|
> |MIS|0.007 ± 0.032|0.006 ± 0.026|0.012 ± 0.064|0.004 ± 0.014|
> |MVC|0.031 ± 0.071|0.016 ± 0.043|0.013 ± 0.033|0.024 ± 0.051|
>
> No systematic degradation, consistent with the TV-distance guarantee.
>
> ### Incomplete experimental analysis. No ablation on predictor architecture, no sensitivity analysis on calibration set size c.
>
> Baseline experiments (relaxed MIPGap, heuristic CP, LSTM vs. linear ablation) are in our response to Reviewer Ph6W; heuristic comparisons in our response to Reviewer Z3pD.
>
> For calibration set size, we sample 100 independent calibration sets of each size and report on the fixed test set.
>
> **Coverage rate (%) by calibration set size**
>
> |Family|c=5|c=10|c=20|c=100|
> |------|-----------|-----------|----------|----------|
> |CFLP|80.0 ± 16.4|87.9 ± 9.1|88.8 ± 8.6|92.7 ± 2.7|
> |GISP|75.1 ± 18.2|80.5 ± 12.0|89.7 ± 7.4|92.8 ± 3.3|
> |MIS|77.5 ± 12.5|86.8 ± 5.6|91.4 ± 5.5|95.3 ± 3.0|
> |MVC|77.7 ± 15.4|85.3 ± 11.3|90.5 ± 7.5|96.0 ± 1.5|
>
> Coverage degrades with smaller $c$ but remains reasonable at $c=20$. The method becomes more conservative (longer solve times, lower suboptimality) as $c$ decreases.
>
> ### Comparison with stronger practical baselines
>
> Full MIPGap tables are in our response to Reviewer Ph6W. In summary: tight gaps (≤0.5%) with comparable quality require 5–14× longer solve times on hard families; loose gaps (5–10%) are faster but incur up to 6.4% suboptimality. Our method is the only one providing a distributional coverage guarantee.
>
> ### CP COPT underperformance across multiple families
>
> The empirical correct rate falls somewhat below 95% for some families in CP COPT, which is possible given the guarantees that conformal prediction provides. Specifically, the guarantees are marginal (averaged over test-set draws) and can fall below the threshold for a specific test set.
>
> In addition, the limited gains of CP COPT over its non-CP version come from a structural difference: COPT tends to find the optimal incumbent later, when the certified gap is already low (e.g., 0.3–2.1% for COPT vs. 0.8–4.7% for Gurobi on MVC/OTS/MMCN), leaving less room for early stopping gains.
>
> ### Positioning relative to (Clarke and Stellato, 2025).
>
> That work differs from the current paper in that we learn a stopping criterion for the exact solver rather than validating an independent heuristic. In that work, conformal prediction bounds the error of a learned solution; here we calibrate a stopping criterion with explicit error probability and magnitude guarantees.
>
> ### Below several works you can refer to.
>
> Thank you. We have added all six references. Han et al. (2023) and Liu et al. (2024, L2P-MIP) are cited in our primal heuristics and solver configuration discussions; Wang et al. (2023) was already cited. The closest is Apollo-MILP (Liu et al., 2025), which uses uncertainty bounds on predicted variables to iteratively reduce problem size. However, Apollo-MILP targets primal solution quality with no guarantee on when to stop an exact solver. Our method is complementary and could be applied on top of it.

---

> > ### Author Rebuttal · Reviewer_1FQN · 2026-04-02
> >
> > My concerns have been addressed. I would like to raise the score.

---

### Official Review · Reviewer_Ph6W · 2026-03-13

**Soundness:** 2
**Presentation:** 4
**Significance:** 2
**Originality:** 2
**Overall Recommendation:** 4
**Confidence:** 4

**Summary:**

The authors analyze a central concept in accelerating mixed-integer programming (MIP) solvers by learning when to terminate branch-and-bound early rather than speeding up the search for optimality itself. This submission explores the concept of training a lightweight neural network to predict the true optimality gap from the solver state and then using conformal prediction (CP) to calibrate a stopping threshold with finite-sample probabilistic guarantees.

**Compliance With Llm Reviewing Policy:**

Affirmed.

**Final Justification:**

The authors provide detailed response and solved my concerns. The score is raised from 3 to 4.

**Key Questions For Authors:**

see "Strengths And Weaknesses"

**Limitations:**

see "Strengths And Weaknesses"

**Strengths And Weaknesses:**

# Strengths

The paper is exceptionally well written, with elegant theory (Theorems 4.1 and 5.1–5.2 reduce to standard exchangeability + monotonicity arguments) that is simple and easy to follow.

---

# Weaknesses

**Limited testbed:** All experiments are performed on Distributional MIPLIB. These families consist of highly homogeneous instances that share identical mathematical structure (same number and type of variables/constraints, only the cost vector or right-hand sides are sampled via θ). Therefore, the convergence curves (upper/lower bounds over time and nodes) might be statistically very similar. It would substantially strengthen the paper if the authors visualized representative convergence curves for several instances per family and explicitly quantified their similarity. Without such visualization, it remains unclear how much of the reported performance is due to the method versus the artificial homogeneity of the benchmark. Standard MIPLIB, by contrast, contains heterogeneous instances with wildly different curve shapes; the method is not tested there.

**Extreme simplicity of the predictor:** The LSTM receives only scalar inputs at each callback (current U(t)/L(t), their 1/3/5-second rolling averages, elapsed time t, and number of explored nodes) and outputs a scalar estimate of the true gap. No problem structure, no variable values, and no full MIP formulation are ever seen. That such a lightweight model works well actually reinforces the hypothesis that convergence curves within each family are highly similar. For example, a straightforward hand-crafted rule—“terminate when the 5-second rolling rate of change of the lower bound falls below a family-specific threshold (or when the upper bound has plateaued for 30 seconds)”—could perfectly capture the “flattening” signal that the LSTM is learning. One could simply plug this naive heuristic into the same framework and obtain identical probabilistic coverage without any neural network. The authors do not ablate the LSTM or compare against such a baseline, leaving open the question of whether the learned component is necessary.

**Missing comparison to relaxed MIPGap:** The only baselines in Table 2 and Appendix A are the default ε-optimality termination of Gurobi/COPT ($\epsilon$ = 0.001) and the trivial “stop after 1 or 3 feasible solutions.” The authors never compare against the standard practitioner approach to early stopping: simply relaxing the solver’s target optimality gap (e.g., setting Gurobi’s MIPGap parameter to 1%, 2%, 5%, or 10%). Given the extreme homogeneity of Distributional MIPLIB, it is highly likely that the LSTM has simply learned this empirical correlation (“stop the solver when the solver gap reaches X%” where X is greater than 0.001). In other words, **the proposed method might merely be acting as a proxy for running the baseline solver with a larger $\epsilon$ tolerance**. The paper would be much stronger if the authors added this direct comparison (and reported results for multiple target ε values beyond the single 0.1% setting).

**Impact:** In fact, the hardest challenge in MIP solver development is precisely diversity. Different instances can exhibit radically different convergence curves. For some instances early stopping works beautifully because the bulk of runtime is spent proving optimality (lower bound tightening via cuts while the upper bound has already stabilized). However, for substantially many instances (especially on heterogeneous benchmarks like standard MIPLIB), the bottleneck is finding high-quality feasible solutions in the first place: lower bounds rise quickly via cutting planes, while upper bounds improve slowly. A method that shines only on parametric families where curves are statistically interchangeable may therefore have limited impact on real-world MIP practice.

---

> ### Author Rebuttal · Authors · 2026-03-31
>
> We thank the reviewer for the detailed and constructive feedback and for appreciating the quality of our writing and the theoretical results from our work.
> This feedback is very helpful to us, and we have done our best to use it to improve the paper.
>
> ### Convergence curves might be statistically very similar. It would strengthen the paper if the authors visualized representative convergence curves and quantified their similarity.
>
> We have added convergence curve analysis to the paper. The curves exhibit substantial heterogeneity:
>
> |Family|UB conv. (s)|LB conv. (s)|Certified gap at UB found (%)|Predicted gap at UB found (%)|
> |------|------------|-------------|-----------------------------|-----------------------------|
> |CFLP|6.8 ± 7.3|15.6 ± 40.8|0.108 ± 0.069|0.034 ± 0.022|
> |GISP|19.0 ± 13.0|37.0 ± 8.9|20.3 ± 10.3|0.845 ± 1.791|
> |MIS|10.8 ± 8.8|221.0 ± 209.4|1.85 ± 0.68|0.014 ± 0.037|
> |MMCN|61.4 ± 110.7|245.2 ± 230.7|2.67 ± 1.88|0.149 ± 0.177|
> |MVC|9.7 ± 7.8|126.7 ± 188.1|0.70 ± 0.18|0.009 ± 0.020|
> |OTS|84.2 ± 18.4|431.8 ± 139.7|4.09 ± 1.66|0.018 ± 0.071|
>
> The LB/UB convergence time ratio ranges from 1.25x (OTS) to 19x (MIS), explaining the speedup variation. The certified gap at UB found varies by two orders of magnitude (0.1% to 20%). Crucially, the LSTM-predicted gap is 3-100x smaller than the certified gap, demonstrating the predictor adds real signal beyond what the solver certifies.
>
> ### That such a lightweight model works well reinforces the hypothesis that convergence curves are highly similar. A straightforward hand-crafted rule could capture the "flattening" signal. The authors do not ablate the LSTM or compare against a hand-crafted baseline.
>
> The main goal of this paper is to highlight how conformal prediction helps terminate MIP solvers. As such, any heuristic could be combined, including the one that the reviewer presented. We include an LSTM vs. linear model ablation and the suggested hand-crafted rule (CP UB rate).
>
> **Solve time (s):**
>
> |Family|LSTM (ours)|Linear|CP UB rate|Full solve|
> |------|------------|-------------|-------------|-------------|
> |CFLP|3.7 ± 4.2|9.4 ± 26.8|4.9 ± 1.1|15.6 ± 40.8|
> |GISP|34.0 ± 9.4|38.4 ± 9.1|34.4 ± 8.0|37.0 ± 8.9|
> |MIS|11.7 ± 5.9|215.4 ± 206.6|12.6 ± 3.1|221.0 ± 209.4|
> |MMCN|83.9 ± 124.9|244.8 ± 229.3|235.7 ± 231.2|245.2 ± 230.7|
> |MVC|14.3 ± 28.8|125.3 ± 186.7|8.5 ± 5.2|126.7 ± 188.1|
> |OTS|96.2 ± 9.2|373.9 ± 154.2|261.5 ± 90.1|431.8 ± 139.7|
>
> The linear model and CP UB rate are competitive on easy families but can be slower on hard ones (e.g., linear: 373s vs LSTM: 96s on OTS; CP UB rate: 128 vs LSTM: 96s on MIS, and almost no speedup on MMCN). All methods achieve low suboptimality (≤0.08%). The below table shows the proportion of the dataset on which the LSTM beats CP UB rate:
>
> |Problem|LSTM beats ub_change (%)|
> |---|---|
> | CFLP-medium | 83/100 (83.0%) |
> | GISP-easy | 47/100 (47.0%) |
> | MIS-medium | 76/100 (76.0%) |
> | MMCN-medium-BI | 80/100 (80.0%) |
> | MVC-medium | 7/100 (7.0%) |
> | OTS-medium | 28/28 (100.0%) |
>
> We have highlighted these comparisons in the paper.
>
>
> ### The proposed method might merely be acting as a proxy for running the solver with a larger ε tolerance.
>
> Results for MIPGap=0.1% are in the paper. No fixed gap performs well across all families.
>
> **Solve time (s):**
>
> |Family|LSTM (ours)|1%|2%|5%|10%|
> |------|-----------|-----|-----|----|----|
> |CFLP|3.7|1.0|0.4|0.2|0.2|
> |GISP|34.0|44.9|44.2|44.1|39.7|
> |MIS|11.7|144.4|33.2|1.6|1.1|
> |MMCN|83.9|213.3|105.7|6.9|2.1|
> |MVC|14.3|33.8|6.2|2.6|0.4|
> |OTS|96.2|137.8|133.5|98.7|74.7|
>
> **Relative suboptimality (%):**
>
> |Family|LSTM (ours)|1%|2%|5%|10%|
> |------|-----------|-----|-----|-----|-----|
> |CFLP|0.015|0.408|0.918|2.069|6.427|
> |GISP|0.011|0.000|0.000|0.046|0.102|
> |MIS|0.004|0.014|0.178|1.436|3.288|
> |MMCN|0.018|0.003|0.029|0.608|1.544|
> |MVC|0.032|0.167|0.484|1.997|5.362|
> |OTS|0.007|0.000|0.000|0.075|2.634|
>
> Tight gaps (≤1%) with comparable quality require much longer solve times. Loose gaps (5–10%) are faster but incur up to 6.4% suboptimality. Our method achieves 0.004–0.032% suboptimality at solve times matching MIPGap=2–5%, plus a distributional coverage guarantee that fixed gaps cannot provide.
>
> ### The bottleneck is finding high-quality feasible solutions in the first place.
>
> In many realistic scenarios lower bounds take very long to converge (see first table). Commercial MIP solvers add cutting planes at the root node and do not use them later, which limits lower bound progress. Our method is complementary: combining a learned primal heuristic with our conformal stopping time would provide both fast solutions and rigorous guarantees (future work direction).

---

> > ### Author Rebuttal · Reviewer_Ph6W · 2026-04-03
> >
> > The authors provide detailed response and solved my concerns. The score is raised from 3 to 4.

---

### Official Review · Reviewer_PbgW · 2026-03-16

**Soundness:** 2
**Presentation:** 3
**Significance:** 3
**Originality:** 2
**Overall Recommendation:** 3
**Confidence:** 3

**Summary:**

This paper addresses the inefficiency of mixed-integer optimization solvers, which often spend substantial time proving optimality after finding a near-optimal solution. To tackle this, the authors propose a method that combines neural prediction with conformal prediction to enable rigorous early stopping. The approach trains an LSTM to estimate the true optimality gap from the solver’s real-time state then uses conformal calibration to set a stopping threshold that guarantees ϵ-optimal solutions with a pre-specified probability (1-α). Experiments on six problem families from Distributional MIPLIB show that the method reduces solve time by over 60% on most benchmarks while maintaining 0.1%-optimality with 95% probability, outperforming heuristic stopping rules and baseline solvers. Overall, the work offers a practical framework for time-critical MIO applications, bridging heuristic intuition and formal reliability.

**Compliance With Llm Reviewing Policy:**

Affirmed.

**Key Questions For Authors:**

See weakness

**Strengths And Weaknesses:**

*Strengths*
1. The method is technically rigorous, with solid theoretical foundations: the conformal prediction framework provides distribution-free, finite-sample guarantees, and supplementary theorems establish generalization of empirical performance to new instances. Experiments are well-designed, covering diverse problem types, multiple solvers, and clear baselines, with detailed tabular and visual results validating speedup and optimality. The LSTM predictor, though simple, is appropriately tailored to the sequential solver state data.
2. The paper is clearly structured, with a logical flow from problem motivation (optimality proof inefficiency) to method design (prediction + conformal calibration) and experiments. It effectively contextualizes the work against traditional MIP acceleration and learning-based optimization, highlighting the unique value of probabilistic guarantees. Technical details on the predictor, calibration procedure, and theoretical proofs are well-organized, making the core logic accessible even to readers less familiar with conformal prediction.
3. While “predicting stopping points” and LSTM-based sequential prediction are not novel, the paper’s core contribution lies in the creative integration of conformal prediction with MIP early stopping. This combination transforms heuristic stopping into a rigorous procedure with quantifiable reliability, a novel angle for MIP acceleration. The work also provides sample-conditional theoretical guarantees that strengthen confidence in real-world deployment, distinguishing it from ad-hoc learning-based approaches.

*Weaknesses*
1. The method’s performance is highly dependent on the similarity between training/calibration instances and test instances—distribution shift (e.g., unseen problem types or scales) is not adequately evaluated, raising concerns about generalization beyond the tested benchmarks. Additionally, the training process requires solving numerous instances to record solver states, which is computationally expensive and limits applicability to problems where existing solvers can already generate sufficient training data.
2. The paper overemphasizes the novelty of “early stopping via prediction” without fully acknowledging prior heuristic methods, leading to a somewhat inflated sense of innovation. The discussion of why LSTM is chosen over more modern sequential models (e.g., Transformers) or graph-based models (common in MIP learning) is insufficient, leaving readers questioning if model choice is arbitrary or task-justified.
3. Impact is constrained by the method’s inability to extend the frontier of tractable MIP instances. Speedup also varies drastically by problem family (e.g., 8% for GISP vs. 95% for MIS), indicating the approach’s effectiveness is highly problem-dependent, limiting its broad utility.

---

> ### Author Rebuttal · Authors · 2026-03-31
>
> We thank the reviewer for the comments and recognition of our method’s technical rigor and the novelty of integrating conformal prediction with MIP early stopping.
> The feedback is very helpful and we believe we can use it to further strengthen our paper.
>
> ### The method’s performance is highly dependent on the similarity between training/calibration instances and test instances. Additionally, the training process requires solving numerous instances to record solver states, which is computationally expensive and limits applicability to problems where existing solvers can already generate sufficient training data.
>
> We do not claim our method solves all MIPs; for applications involving repeated solves, it reduces computational cost. This assumption is standard in the learning-for-MIP literature. **In fact, our work requires substantially less training data than most other recent works:**
>
> |Paper|Authors|Year|Training Instances|Citations|
> |-------------------------------------------|-------------|--------|------------------|---------|
> |Towards Foundation Models for MIP|Li et al.|2024|38,256|39|
> |Learning Cut Selection for MILP|Wang et al.|2023|10,000|91|
> |Hybrid Models for Learning to Branch|Gupta et al.|2020|10,000|227|
> |Learning to Branch in MIP|Khalil et al.|2016|~10,000|411|
> |Exact Combinatorial Optimization with GCNNs|Gasse et al.|2019|10,000|601|
> |**Our paper**|**Anonymous**|**2026**|**up to 1,000**|**—**|
>
> The small data requirement is a strength of our work.
>
> We have also added theory and experiments on the effect of distribution shift in the test instances, and we include the details in our response to Reviewer 1FQN, who asked specifically about this.
>
> ### The discussion of why LSTM is chosen over more modern sequential models (e.g., Transformers) or graph-based models (common in MIP learning) is insufficient.
>
> We address this in two parts.
>
> #### 1. Why not use a simpler model, such as a linear model?
>
> We have included an ablation study substituting the LSTM for a linear model and the heuristic suggested by reviewer Ph6W. The results of this study are included in our response to Reviewer Ph6W.
>
> #### 2. Why an LSTM rather than a more complex, modern model like a transformer or graph neural network?
>
> The aim of this paper is to demonstrate that it is possible to solve mixed integer optimization problems with statistical guarantees on optimality using a learned estimate of the lower-bound combined with conformal prediction.
> The main novelty is in the rigorous optimality bounds given by the conformal predictor, which are not present in previous works.
> It is true that in some applications, the optimization problem might be sufficiently hard, or the solve might take sufficiently long, that a simple RNN might not perform well. In this case, practitioners may want to swap this component out for a transformer, or include a graph neural network to add awareness of the entire problem formulation. We added a discussion about this in the paper.
>
> ### Impact is constrained by the method’s inability to extend the frontier of tractable MIP instances. Speedup also varies drastically by problem family (e.g., 8% for GISP vs. 95% for MIS).
>
> This is true of almost all learning-for-MIP research; all works in the above table carry the same assumption.
>
> Reducing time spent in online solves of tractable but costly MIPs can have large impact: in high-frequency trading, faster portfolio optimization gives an edge over competitors; in model-predictive control, early stopping saves energy on repeated trajectory solves while our guarantees ensure feasibility and near-optimality.
>
> Our method performs very well on some families (95% speedup on MIS) and offers modest improvements on others. It would be unrealistic to expect 95%+ speedups on all parametric families. We note that speedup variability across problem families is characteristic of the learning-for-MIP literature. Among the works listed above, Khalil et al. (2016) report 29% and 14% speedups on medium and hard instances but are 57% slower on easy instances and 10% slower overall due to per-node feature computation overhead. Gasse et al. (2019) achieve 11–41% speedups on three families but are 74% slower on Maximum Independent Set (medium). Gupta et al. (2020) report 24–47% speedups on medium instances but only 2–6% on big instances. Even modest speedups (8–10%) are meaningful in settings where solves are repeated thousands of times, such as portfolio optimization or model-predictive control.

---

> > ### Author Rebuttal · Reviewer_PbgW · 2026-04-04
> >
> > Thanks for the reply. My concerns are addresses except for the early stopping part (weakness2), could you please explain it?

---

> > > ### Author Response · Authors · 2026-04-04
> > >
> > > Thank you for the follow-up. We agree: the idea of stopping early is not new. Practitioners already use relaxed MIPGap tolerances, time limits, or solution-count thresholds. What is new is the guarantee: we can stop early and certify that the returned solution is $\epsilon$-optimal with probability at least $1-\alpha$. No prior early stopping method provides this. To see why it matters concretely, consider our MIPGap comparison (added in the revision): relaxed MIPGap=5% reaches similar solve times to our method but incurs up to 6.4% actual suboptimality, while our method achieves 0.004–0.032% suboptimality with a coverage guarantee. The gap tolerance gives a worst-case bound; our method gives a distributional one, which is tighter in practice. We have clarified this distinction in the revised paper.

---

### Decision · Program_Chairs · 2026-04-30

**Decision:**

Accept (regular)

**Comment:**

This paper introduces a conformal prediction framework for early stopping in mixed-integer optimization, offering finite-sample probabilistic guarantees together with substantial speedups.
Reviewers generally agreed on the novelty, technical soundness, and practical relevance of the approach. The authors' rebuttal addressed most of the main concerns, including questions about dependence on training/calibration-test similarity, the homogeneity of the MIPLIB-related datasets, the use of LSTM as the predictor, the comparisons with heuristic baselines, and the clarification of gap definitions and experimental settings. The discussion overall moved in a positive direction, with at least one reviewer explicitly raising their score, while the remaining weak-reject review did not provide further objections after the authors supplied additional evidence. Therefore, I recommend accepting this paper.